# What's Wrong With Non-Autoregressive Graph Neural Networks in Neural Combinatorial Optimization?

## Abstract

Neural combinatorial optimization (NCO) leverages machine learning models to tackle complex combinatorial problems by learning heuristics or direct solution construction. Graph Neural Networks (GNNs) are particularly effective for NCO due to their ability to capture the relational structure inherent in many such problems. In this work, we examine the supervised non-autoregressive (NAR) solution construction framework, revealing a misalignment between training objective and solution quality. Specifically, through experiments on six GNN architectures across three problems—Traveling Salesperson Problem (TSP), Maximum Independent Set (MIS), and Minimum Vertex Cover (MVC)—we show that lower training loss does not correlate with lower optimality gap. To address this, we propose a supervised autoregressive (AR) framework that leverages the conditional dependencies between variables by training to complete partial solutions. Empirical results show that the proposed AR framework does not exhibit the same misalignment and consistently improves performance. We further compare the proposed AR framework against existing supervised GNN-based methods and achieve superior performance, especially in terms of generalizing to larger problem instances.

## 1 Introduction

Combinatorial optimization problems are fundamental to a wide range of industries, including vehicle routing (Tassone & Choudhury, 2020), machine scheduling (Brucker, 2007), and resource allocation (Xiao et al., 2012). As these problems scale, they become increasingly difficult to solve optimally, presenting significant challenges for exact methods. Despite decades of research, traditional approaches often fall short of addressing the large-scale demands posed by real-world applications, especially given the complexity driven by recent globalization and technological advancements (Bertsimas & Dunn, 2019).

The rise of modern deep learning techniques has opened new avenues for tackling these problems, leading to the emergence of Neural Combinatorial Optimization (NCO) (Khalil et al., 2022; Gasse et al., 2019; Joshi et al., 2019; Li et al., 2018; Khalil et al., 2017; Sun & Yang, 2024; Luo et al., 2024; Qiu et al., 2022). One dominant NCO approach is the application of Graph Neural Networks (GNNs) to construct primal solutions. GNNs, which are specialized deep learning architectures for graph-structured data, have gained prominence in NCO due to their inherent suitability for combinatorial optimization as many such problems (e.g. vehicle routing problems) are naturally represented as graphs. Furthermore, many combinatorial optimization problems can be modeled as constraint-variable graphs (Gasse et al., 2019; Khalil et al., 2022), further emphasizing the alignment between GNNs and the NCO domain.

GNN-based approaches for constructing primal solutions in combinatorial optimization can be broadly categorized based on their underlying learning problem: deep reinforcement learning (Khalil et al., 2017; Qiu et al., 2022; Ahn et al., 2020; Kool et al., 2019; Xing & Tu, 2020; Peng et al., 2020; Zhou et al., 2023), unsupervised learning (Min et al., 2024; Min & Gomes, 2024; Drori et al., 2020; Karalias & Loukas, 2020; Toenshoff et al., 2021; Amizadeh et al., 2019; Wang et al., 2022), and supervised learning (Joshi et al., 2019; Kool et al., 2022; Fu et al., 2021; Sun & Yang, 2024; Li et al., 2024; Huang et al., 2023; Li et al., 2018). This paper primarily focuses on GNN-based supervised learning

methods. These methods are dominated by non-autoregressive (NAR) algorithms where a GNN model, that is trained using optimal solutions as ground-truth labels, assigns independent probabilities to each variable, treating the task as a node or edge classification task. These probabilities, known as a probability map or a heat map, indicate the likelihood of each node or edge being part of the optimal solution. A search algorithm, such as greedy search (Joshi et al., 2019; Sun & Yang, 2024), beam search (Joshi et al., 2019), or Monte-Carlo tree search (MCTS) (Sun & Yang, 2024; Fu et al., 2021), is then applied based on the GNN-generated probability map to construct a valid solution.

While recent supervised approaches have shown state-of-the-art empirical results (Sun & Yang, 2024; Fu et al., 2021), key limitations have been identified in some of these methods, particularly regarding the practicality of the generated probability maps (Li et al., 2018; Xia et al., 2024) and their ability to scale to larger problem instances (Fu et al., 2021; Joshi et al., 2022). To better understand these limitations, we conduct an extensive experimental evaluation to analze the connection between the quality of the generated probability maps and the quality of the solutions they produced. Our analysis revealed a notable misalignment: improvements in the quality of probability maps do not correlate with higher quality of constructed solutions. We hypothesize that the non-autoregressive nature of these approaches is the source of the misalignment and propose to use autoregressive models to resolve the misalignment. Extensive experiments provide empirical support to our hypothesis and demonstrate how the use of autoregressive models mitigate the observed misalignment and improve the performance of these models.

Our contributions are summarized as follows:

1. We examine the supervised NAR solution construction framework, that encompass several notable approaches, and identify a clear misalignment between the accuracy of the probability maps (i.e., the training objective) and the quality of the constructed solutions. Specifically, we run experiments for six different GNN architectures across three different graph-based combinatorial optimization problems— Traveling Salesperson Problem (TSP), Maximum Independent Set (MIS), and Minimum Vertex Cover (MVC)— and show that improvements in training loss do not correlate with improvements in optimality gap.

2. We introduce a general framework for supervised AR solution construction leveraging conditional generation by training to complete randomly sampled partial solutions. Our results show that this framework, tested on the same three combinatorial optimization problems and six GNN architectures, does not exhibit the previously observed misalignment and consistently leads to higher quality primal solutions.

3. We compare the proposed AR framework against existing supervised GNN-based methods on TSP instances of various sizes. Our experimental results show that our AR framework achieves superior performance, especially in terms of generalizing to larger instances.

## 2 BACKGROUND

In this section, we give some background on the general framework of GNN-based supervised methodologies. Formally, given an instance $g$ of combinatorial optimization problem $\mathcal{G}$ with binary variables $D_g$, we denote $\mathcal{X}_g \subseteq 2^{D_g}$ as the set of feasible solutions of $g$ and $c_{\mathcal{G}} : \mathcal{X}_g \to \mathbb{R}$ as the objective function. To goal is to find the optimal solution defined as:

$$\hat{x}_g = \arg \min_{x_g \in \mathcal{X}_g} c_{\mathcal{G}}(x_g) \tag{1}$$

Instead of searching through the large discrete solution space $\mathcal{X}_g$, existing methods define a continuous solution space $\Omega_g \subseteq [0, 1]^{|D_g|}$ and a model $\mathcal{M}$ with parameters $\theta$:

$$\mathcal{M}_\theta : \mathcal{G} \to \Omega_g \tag{2}$$

is tasked with predicting a $|D_g|$-dimensional vector $\omega \in \Omega_g$, representing a probability map where each entry $\omega_i$ estimates the probability of variable $i$ being true in optimal solution $\hat{x}_g$. Then, a search algorithm $S_{\mathcal{G}} : \Omega_g \to \mathcal{X}_g$ constructs a feasible solution $x_g$ by searching through the predicted probability map $\omega$. Therefore, the objective of the model is defined as:

$$\mathcal{L}(\theta) = \mathbb{E}_{g \sim \mathcal{G}} \left[ c_{\mathcal{G}}(S_{\mathcal{G}}(\mathcal{M}_\theta(g))) \right] \tag{3}$$

However, since the objective functions $c_\mathcal{G}$ of combinatorial optimization problems and the search algorithms $S_\mathcal{G}$ are generally non-differentiable (Qiu et al., 2022), this objective cannot be directly optimized. Instead, existing methods adopt some surrogate loss function $l_{\text{surrogate}}$ to approximate $c_\mathcal{G}(S(\cdot))$. Then, the optimization objective becomes:

$$\mathcal{L}'(\theta) = \mathbb{E}_{g \sim \mathcal{G}} \left[ l_{\text{surrogate}}(\mathcal{M}_\theta(g)) \right] \tag{4}$$

Most commonly, a supervised classification loss (e.g. cross-entropy loss) is employed to treat this as a classification task (Joshi et al., 2019; Sun & Yang, 2024; Luo et al., 2024; Vinyals et al., 2015). That is, we can construct a training set by finding the optimal solution $\hat{x}_g$ for problem instance $g$ and using $\hat{x}_g$ as the ground-truth labels for the variables (1 for true in $\hat{x}_g$ and 0 otherwise). Then, we can train $\mathcal{M}$ as a binary classification problem. A GNN model is a common choice for $\mathcal{M}$ as many combinatorial optimization problems are naturally defined on graphs (Joshi et al., 2019; Sun & Yang, 2024; Fu et al., 2021; Kool et al., 2022). For example, the TSP and many related vehicle-routing problems can be defined on a graph where nodes represent locations, edges represent routes, and edge weights represent costs. Following this setup, the surrogate objective in Eq. 4 becomes a binary node/edge-level classification task.

This general framework is non-autoregressive (NAR) in nature. That is, the probability outputs of each variable are generated independent of each other (Sun & Yang, 2024). On the other hand, in autoregressive (AR) approaches, the model would generate predictions conditioned on its previous predictions. In this context, models would generate probabilities iteratively, where each output is conditioned on previous outputs (Luo et al., 2024). However, to the best of our knowledge, there is no existing supervised GNN-based AR method.

## 2.1 RELATED WORK

**NAR Methods.** Following the general NAR framework, notable works include Joshi et al. (2019) who approach the TSP as a binary edge classification task, using the optimal TSP tour as labels (in this work, we term this approach EFFICIENTTSP for brevity). The model's backbone is a Residual Gated Graph Convolutional Network (GatedGCN) (Bresson & Laurent, 2017), trained using cross-entropy loss. To construct a valid TSP tour based on the generated probability map, EFFICIENTTSP employs a greedy search algorithm, which can also be generalized to a beam search. Another notable work is Li et al. (2018) who target the MIS problem and, using optimal solutions as binary node labels, train a GNN model to generate multiple probability maps per instance. A tree search method is deployed to search through these probability maps to construct a solution.

Subsequent works have expanded on this general framework by incorporating novel architectures or complex search algorithms. For instance, Fu et al. (2021) train a GNN model to generate probability maps for subgraphs of a TSP instance, which are then merged into a comprehensive probability map used by a Monte Carlo Tree Search (MCTS) algorithm to construct a valid tour. Additionally, Kool et al. (2022) introduced Deep Policy Dynamic Programming (DPDP) which uses dynamic programming as the search algorithm to construct valid solutions for TSP and related routing problems. Recently, Sun & Yang (2024) proposed DIFUSCO, a method that integrates diffusion-based GNN models within the existing framework, achieving state-of-the-art results on TSP, with further post-processing techniques presented by Li et al. (2024) and Huang et al. (2023).

Recent research has highlighted limitations in the existing NAR methods. Notably, Joshi et al. (2022) found that these methods exhibit poor generalization capabilities. Additionally, concerns have been raised regarding the practicality of these generated probability maps when decoded by complex post-hoc search algorithms. In particular, Xia et al. (2024) demonstrated that a simple softmax-based heuristic could produce probability maps yielding solutions of comparable quality to those produced by trained GNNs when decoded with MCTS. Similarly, Böther et al. (2022) showed that the guided tree search method in (Li et al., 2018) can construct near-optimal solutions even from random probability maps.

However, these studies are limited to tree search methods and do not investigate the link between the quality of the probability maps and the quality of the resulting solutions. To our knowledge, our study is the first to identify the misalignment between probability map quality and solution quality in supervised NAR approaches.

**AR Methods.** Despite its absence within the supervised GNN domain, we note that AR methods have been proposed in NCO through other learning paradigms, such as deep reinforcement learning (Kool et al., 2019; Qiu et al., 2022), and in other architectures like pointer networks (Vinyals et al., 2015) and transformers (Bresson & Laurent, 2021; Luo et al., 2024). We also note that Li et al. (2018)'s MIS-specific graph reduction technique behaves similarly in concept to an AR approach. Though, it is specific to the MIS problem and is not compatible with other types of problems. We further discuss this in Appendix C.

## 3 THE MISALIGNMENT IN THE GENERAL SUPERVISED NAR FRAMEWORK

In this section, we conduct experimental analysis of the NAR framework to understand the observed limitations. Specifically, we implement and evaluate the existing NAR framework by continuously tracking and comparing the quality of the generated probability maps with the quality of the constructed solutions after each training epoch. This is performed across three NP-complete combinatorial optimization problems (Hartmanis, 1982): TSP, MIS, and MVC. While all three problems are well studied in NCO literature, TSP has been a focal point in NCO research (Sun & Yang, 2024; Joshi et al., 2019; Fu et al., 2021), while MIS (Li et al., 2018; Sun & Yang, 2024) and MVC (Khalil et al., 2022) complement it by representing node-based decision variables, as opposed to the edge-based decision variables in TSP.

TSP involves finding the minimum-cost cycle that visits each node exactly once, MIS involves finding the largest set of non-adjacent nodes in a graph, and MVC involves finding the smallest set of nodes such that at least one endpoint of each edge is included. Formal definitions of these problems are included in Appendix A.

The implemented method follows the workflow outlined in Section 2. We define a training instance as the input graph with labels for the decision variables extracted from the optimal solution (1 if included in the optimal solution, 0 otherwise). A GNN model is trained to predict these labels using cross-entropy as the surrogate loss. This approach mirrors the probability map generation process employed in all aforementioned NAR methods (Joshi et al., 2019; Kool et al., 2022; Fu et al., 2021; Sun & Yang, 2024).

### 3.1 EXPERIMENTAL SETUP

**GNN Architectures.** Our experiments include six representative GNN architectures: Graph Attention Network (GAT) (Veličković et al., 2018; Brody et al., 2022), Residual Gated Graph Convolutional Network (GatedGCN) (Bresson & Laurent, 2017), Graph Convolutional Network (GCN) (Kipf & Welling, 2017), GraphSage (Hamilton et al., 2017), MoNet (Monti et al., 2017), Graph Isomorphism Network (GIN) (Xu et al., 2018). Given DIFUSCO's state-of-the-art performance on TSP, we also included its diffusion-based GNN model, which uses GatedGCN as its backbone, in our evaluation[1]. For a detailed description and review of these architectures, we refer readers to the GNN benchmark paper by Dwivedi et al. (2023).

**Solution Construction.** Due to the concerns raised by Li et al. (2018) and Xia et al. (2024) regarding the practicality of probability maps under complex search algorithms, we employ greedy search in order to evaluate the impact of the probability maps in isolation. The greedy search algorithm involves repeatedly adding the variable with the highest probability to the solution set without violating problem-specific constraints (i.e. no visiting nodes twice for TSP, no adjacent nodes for MIS, and none for MVC) until some problem-specific termination condition (i.e., a hamiltionion cycle for TSP, no more independent nodes for MIS, and all edges are covered for MVC). Furthermore, for TSP, we also enforce that the edges are added sequentially, maintaining a connected path at all times, following the convention of EFFICIENTTSP and DIFUSCO (Joshi et al., 2019; Sun & Yang, 2024). Details on the exact decoding process for each problem are provided in Appendix E.

---

[1]Due to technical difficulties running the publicly available DIFUSCO implementation, we only report the results for DIFUSCO on TSP.

**Metrics.** We use optimality gap as the primary metric for solution quality, defined as:

$$\text{Optimality Gap} = \frac{|z_{\text{pred}} - z_{\text{opt}}|}{z_{\text{pred}}} \times 100\% \tag{5}$$

where $z_{\text{pred}}$ and $z_{\text{opt}}$ represent the objective values of the constructed solution and the optimal solution, respectively. The quality of the generated probability maps is evaluated using the cross-entropy loss metric. In this set of experiments, we focus on the misalignment in training of models and therefore only evaluate these metrics for instances in the training set. Evaluation based on a held-out validation or test set could conflate the misalignment between training loss and optimality gap with misalignment between the training and testing distributions. However, in Section 5 we demonstrate the impact of this misalignment on the performance of NAR approaches on a held-out test set.

**Dataset.** For each problem, the training set consists of 5,000 random synthetically generated problem instances, each with 100 nodes. The ground-truth labels are generated using Gurobi (Gurobi Optimization, LLC, 2023). Details on the problem generation methods are provided in Appendix D.

**Configurations.** For each model, we use a default configuration of 4 message-passing layers with 128 hidden dimensions. A prediction head consisting of 2 fully connected layers is applied to the hidden state of each decision variable. We use the Adam optimizer (Kingma & Ba, 2014) with a decaying learning rate initialized at 0.001. These configurations follow the general conventions found in a previous work by Joshi et al. (2022). The exact hyperparameters for each architecture can be found in Appendix F. For the diffusion model, we also use 4 message-passing layers with 128 hidden dimensions, but otherwise follow the implementation of DIFUSCO[2] (Sun & Yang, 2024).

**Hardware** All experiments were conducted on an NVIDIA GeForce RTX 4080 Ti.

## 3.2 EXPERIMENTAL RESULTS

Figure 1 shows both the training loss and the optimality gap across training epochs for the various GNN architectures. In general, we observe that the training losses (dotted lines) consistently decrease throughout the training process, indicating improvement in the quality of the probability maps. However, the optimality gaps (solid lines) do not follow the same trend, often oscillating instead. Notably, in many cases, the optimality gaps do not improve beyond the first epoch.

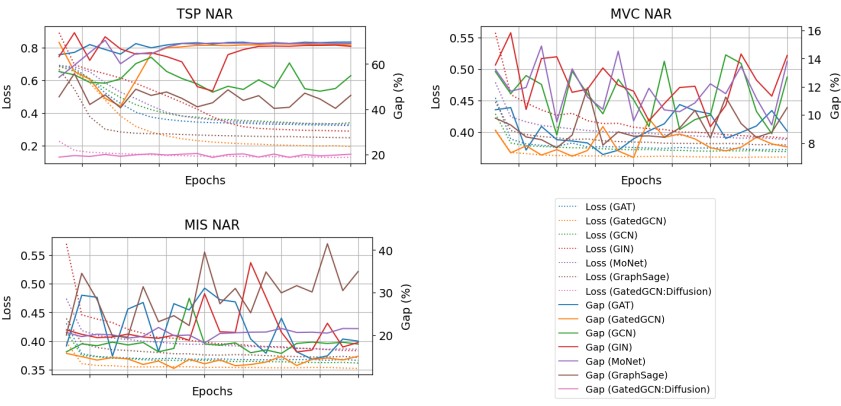

Figure 1: Results of the general NAR framework evaluated on TSP, MVC, and MIS across six different GNN architectures, comparing training loss (dotted lines) and optimality gap (solid lines) throughout the training phase. Lower is better for both metrics.

These results indicate that, under the NAR framework, a decrease in training loss does not correlate with a decrease in optimality gap. This misalignment between the quality of probability maps and the quality of constructed solutions is observed consistently across all problem types and all architectures.

---

[2]https://github.com/Edward-Sun/DIFUSCO

This points to a fundamental issue within the general supervised NAR framework where the surrogate loss function (Eq. 4) does not accurately represent the desired optimization objective (Eq. 3).

We also performed this evaluation on the EFFICIENTTSP and DIFUSCO models following their original configurations, fully replicating the experimental setup described in their respective manuscripts (Joshi et al., 2019; Sun & Yang, 2024). We also included a version of the EFFICIENTTSP model trained on 1,000,000 problem instances (instead of 10,000) per epoch for completeness. As shown in Figure 2, the results are consistent with those presented in Figure 1, showing lack of correlation between optimality gap and loss throughout the training phase.

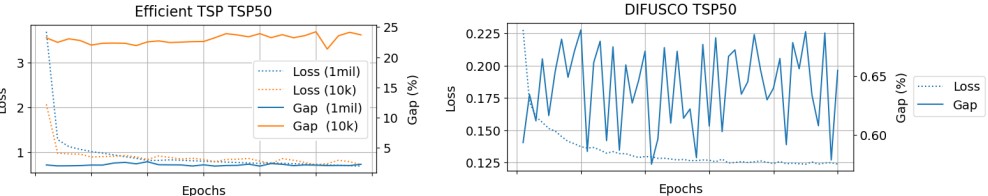

Figure 2: Results of EFFICIENTTSP (Joshi et al., 2019) and DIFUSCO (Sun & Yang, 2024), following their original configurations, comparing training loss (dotted lines) and optimality gap (solid lines) throughout the training phase. For EFFICIENTTSP, the value in brackets indicates the number of problem instances used for training. Lower is better for both metrics.

We hypothesize that the misalignment between the quality of the probability maps and the constructed solutions may be due to the non-autoregressive nature of the model, where conditional dependencies are not captured. This results in the selection of nodes or edges that do not account for or adapt to the choices made earlier in the construction process, potentially leading to suboptimal solutions.

## 4 AUTOREGRESSIVE FRAMEWORK FOR GNN-BASED NCO

In this section, we present a supervised GNN-based AR solution construction framework. The motivation for the proposed approach stems from the misalignment identified in Section 3 between the desired optimization objective defined in Eq. 3 and the surrogate loss function defined in Eq. 4. The proposed method addresses this issue by taking an AR approach and changing the optimization objective of the model accordingly. Specifically, we propose a novel optimization objective from an AR perspective that aims to construct a valid solution by iteratively generating conditional probability maps after each variable selection.

Formally, given a binary combinatorial optimization problem instance $g \in \mathcal{G}$ with variable set $D_g$, feasible solution set $\mathcal{X}_g \subseteq 2^{D_g}$, and objective function $c_{\mathcal{G}} : \mathcal{X}_g \to \mathbb{R}$; we aim to find optimal solution set $\hat{x}_g \in \mathcal{X}_g$ as defined in Eq. 1. However, due to its size, it is infeasible to directly search through the discrete solution space $\mathcal{X}_g$. As such, the proposed method defines a GNN model $\mathcal{M}$ with parameters $\theta$:

$$\mathcal{M}_\theta : (g, \tilde{x}_g) \mapsto \omega \in \Omega_g \tag{6}$$

where $\Omega_g \subseteq [0, 1]^{|D_g|}$ represent probability maps over $D_g$ and $\tilde{x}_g \in \tilde{\mathcal{X}}_g \subseteq 2^{D_g}$ represents a partial solution of $g$. Unlike in the NAR framework (Eq. 2), here the generation of probability maps is *conditioned* on some partial solution $\tilde{x}_g \in \tilde{\mathcal{X}}_g$. To construct a feasible solution, we initialize a partial solution $\tilde{x}_g = \emptyset$. Then, a search algorithm $S_{\mathcal{G}} : \Omega_g \times \tilde{\mathcal{X}}_g \to \tilde{\mathcal{X}}_g$ updates a given partial solution $\tilde{x}_g$ by selecting one variable to be true based on a probablity map $\omega$. This process iterates by generating a new probability map $\omega$ after every update to $\tilde{x}_g$ until some problem-specific termination criteria. This process is autoregressive as each model output $\omega = \mathcal{M}_\theta(g, \tilde{x}_g)$ is conditioned on previous model outputs represented by $\tilde{x}_g$. Following this method, the objective of the model is to predict probability maps that will construct partial solutions with the lowest expected objective value, which is defined as:

$$\mathcal{L}(\theta) = \mathbb{E}_{g \sim \mathcal{G}} \left[ \mathbb{E}_{x_g \sim \mathcal{X}_g | S_{\mathcal{G}}(\mathcal{M}_\theta(g, \tilde{x}_g)) \subseteq x_g} \left[ c_{\mathcal{G}}(x_g) \right] \right] \tag{7}$$

Again, as the objective functions $c_{\mathcal{G}}$ of combinatorial optimizations and search algorithms $S_{\mathcal{G}}$ are generally non-differentiable (Qiu et al., 2022), we approximate $\mathbb{E}_{x_g \sim \mathcal{X}_g | S_{\mathcal{G}}(\mathcal{M}_\theta(g, \tilde{x}_g)) \subseteq x_g} [c_{\mathcal{G}}(x_g)]$ using a classification loss function, $l_{\text{classification}}$, as a surrogate:

$$\mathcal{L}'(\theta) = \mathbb{E}_{g \sim \mathcal{G}} [l_{\text{classification}}(\mathcal{M}_\theta(g, \tilde{x}_g), \hat{x}_g)] \tag{8}$$

Specifically, we use optimal solution $\hat{x}_g \in \hat{\mathcal{X}}_g$ as the ground-truth label and train the model to complete $\hat{x}_g$ from partial solutions $\tilde{x}_g \subset \hat{x}_g$. That is, to approximate $\mathbb{E}_{x_g \sim \mathcal{X}_g | S_{\mathcal{G}}(\mathcal{M}_\theta(g, \tilde{x}_g)) \subseteq x_g} [c_{\mathcal{G}}(x_g)]$, we use the following supervised classification loss:

$$l_{\text{classification}}(\theta \mid g, \tilde{x}_g) = \mathbb{E}_{\hat{x}_g \sim \hat{\mathcal{X}}_g} [l_{\text{CE}}(\mathcal{M}_\theta(g, \tilde{x}_g), \hat{x}_g)] \tag{9}$$

where $l_{\text{CE}}$ is cross-entropy loss. This approach aims to achieve alignment between the surrogate loss function (Eq. 8) with the desired optimization objective (Eq. 7) by significantly simplifying the prediction task. Intuitively, instead of generating a single probability map that would dictate the selection of the entire solution set (as in the NAR framework), the proposed approach is to generate a probability map that is used to select only the next variable to be added to the solution set. This process can then be repeated to construct a valid solution in an autoregressive manner. Not only does the proposed method significantly simplify the desired model objective, it also enables the model to perform conditional generation. That is, the generation of each probability map is conditioned on the current partial solution. Unlike existing NAR methods, the conditional generation in our method allows the model to adjust for any suboptimal choices made earlier in the solution construction process. It also better captures the multimodal nature of combinatorial optimization problems, avoiding the pitfall of getting caught between predicting multiple equally optimal solutions.

## 4.1 TRAINING VIA PARTIAL SOLUTION SAMPLING

To train the model to make predictions conditioned on a partial solution, our method treats the task as a node or edge-level classification task, trained using cross-entropy loss. We construct partial solutions from the ground-truth (i.e., optimal solution) of each training instance and encode them as part of the model input and ask the model to complete each partial solution. In this way, the model is trained to predict the next variable to be included conditioned on a partial solution.

Formally, given problem instance $g \in \mathcal{G}$ and the optimal solution set $\hat{x}_g$, we can construct a partial solution $\tilde{x}_g \subset \hat{x}_g$ and task the model with predicting the remainder $\tilde{x}'_g = \hat{x}_g \backslash \tilde{x}_g$. To explicitly encode the condition on a partial solution $\tilde{x}_g$, each variable in the graph is given a binary feature indicating its inclusion in $\tilde{x}_g$ (1 for included, 0 otherwise). Correspondingly, a binary label is assigned to each variable to indicate whether it should be selected next, with those in $\tilde{x}'_g$ labeled as 1 and others as 0. This setup provides explicit information about the current partial solution and the correct subsequent inclusions, enabling the model to make conditional predictions. We can apply this process many times per problem instance allowing us to significantly increase the diversity and volume of the training set, resulting in a more robust framework, especially in cases where labeled data is scarce.

Ideally, for each labeled problem instance $g$ in the training set and each of its optimal solutions $\hat{x}_g \in \hat{\mathcal{X}}_g$, we construct training data from all possible subsets of $\hat{x}_g$ which would result in training data in the magnitude of $O(2^{|\hat{x}_g|})$ for each optimal solution. For tractability, we instead opt for sampling $k$ partial solutions (and their corresponding labels) from one optimal solution for each problem instance, by first sampling the length of the partial solution uniformly at random and then randomly sampling a subset of the optimal solution of the chosen length. The pseudocode for the proposed framework is provided in Appendix H.

## 4.2 EXPERIMENTAL SETUP

In the same fashion as in Section 3, we conducted experiments on MIS, MVC, and TSP across six different GNN architectures, tracking both training loss and optimality gap throughout the training phase. For each training instance, we sampled $k = 50$ partial solutions of uniformly distributed sizes. We use the same configurations and hardware as outlined in Section 3. The exact hyperparameters are included in Appendix F.

**Solution Construction.**   To construct valid solutions after training, we employed greedy search, as was done previously for the NAR framework. We followed the same greedy process, where

the highest-scored variable is iteratively added, subject to problem-specific constraints. The key difference is that we now generate a new probability map after each iteration. Additionally, for TSP, we no longer enforce that the partial solution be a connected path, ensuring better alignment between training and deployment. During training, the model predicts over all edges, not just those incident to the current partial tour. Therefore, during deployment, we do not require the tour to be constructed sequentially. The decoding process for each problem is described in detail in Appendix E.

## 4.3 EXPERIMENTAL RESULTS

The results of our evaluation, shown in Figure 3, compare training loss and optimality gap for the proposed framework. Across all three problems and all GNN architectures, the optimality gap consistently mirrors the corresponding training loss. In most cases, both metrics exhibit a decreasing trend throughout the training epochs. In some cases both metrics are stagnant, though still consistent with each other. Unlike the NAR framework, the proposed AR framework does not exhibit any evident misalignment between probability map quality and solution quality, indicating a more effective alignment between the model's desired objective and the surrogate objective.

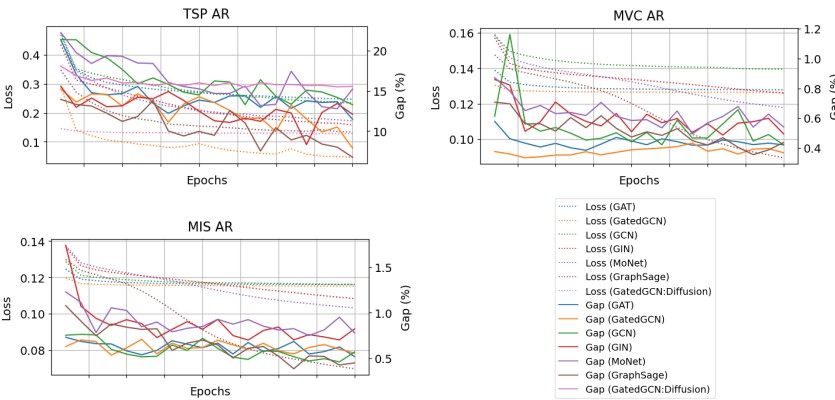

Figure 3: Results of the proposed AR framework evaluated on TSP, MVC, and MIS across six different GNN architectures, comparing training loss (dotted lines) and optimality gap (solid lines) throughout the training phase. Lower is better for both metrics.

As shown in Figure 4, the proposed AR framework achieved better performance across all problems and architectures compared to the NAR framework. In fact, for MIS and MVC, the AR models were able to achieve near optimal solutions within 1% of the optimal objective value. More detailed experiments against existing baselines are conducted in Section 5. Finally, we note that, as expected, the AR method requires multiple inference steps and therefore leads to increase in the total decoding time. For MIS and MVC, on average, the AR models took around two times as long to decode. For TSP, on average, the AR models took around five times as long. Analysis of the differences in runtime is provided in Appendix G.

## 5 COMPARATIVE ANALYSIS

In this section, we conduct experiments comparing the performance of the proposed AR framework against existing supervised GNN-based methods on TSP instances of various sizes. TSP was chosen as the benchmark due to its extensive study within the NCO domain (Joshi et al., 2019; Kool et al., 2022; 2019; Kwon et al., 2020; Sun & Yang, 2024; Khalil et al., 2017; Fu et al., 2021; Min et al., 2024; Luo et al., 2024; Bresson & Laurent, 2021; Deudon et al., 2018).

In our experiments, we first evaluated the models on a test set consisting of random TSP instances that have the same number of nodes as the those in the training set. We also evaluated the models' ability to generalize to larger instances, that is, random TSP instances that have more nodes than those in the training set. We choose to evaluate the models' ability to generalize to larger instances due to the inherent and unique challenges of combinatorial optimization problems that make them

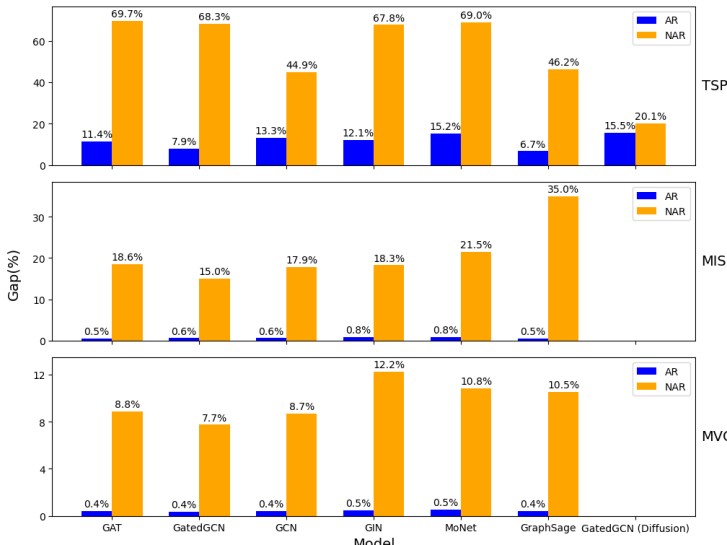

Figure 4: Results comparing the optimality gap (%) achieved by the NAR models and the AR models. Lower is better.

especially challenging at scale. Therefore, it is crucial that NCO methods generalize well to larger instances (Joshi et al., 2022; Fu et al., 2021).

**Baselines.** As the scope of this work is focused on supervised GNN-based methods, we choose EFFICIENTTSP (Joshi et al., 2019) and DIFUSCO (Sun & Yang, 2024) as the baselines for this experiment. All methods are implemented with greedy search as described in previous sections. Other GNN-based supervised methods, such as those by Fu et al. (2021), Kool et al. (2022), and Li et al. (2024), are excluded as they focus on developing complex search algorithms and post-processing which can obfuscate the quality of the models (Xia et al., 2024; Böther et al., 2022). For the hyperparameters of the baselines, please refer to their original manuscripts (Joshi et al., 2019; Sun & Yang, 2024).

**Datasets.** For training, we used 10,000 instances of TSP50[3] for our model. For the baselines, 1,502,000 instances of TSP50 are used to train DIFUSCO and 10,000 instances of TSP50 are used to train EFFICIENTTSP, as per their original manuscripts. We also included a version of the DIFUSCO model trained with 10,000 instances for fair comparison. For validation, we used 1,000 instances of TSP50. For testing, we used 1,000 instances of TSP50, 1,000 instances of TSP100, 200 instances of TSP200, and 50 instances of TSP500. All TSP datasets are generated closely following the convention set by existing works (Joshi et al., 2019; Fu et al., 2021; Min et al., 2024; Sun & Yang, 2024; Luo et al., 2024). For details regarding the problem generation process, please refer to Appendix D.

**Configurations.** We used the GatedGCN architecture (Bresson & Laurent, 2017) with 8 layers and 256 hidden dimensions. We sampled $k = 200$ partial solutions for each TSP instance in the training set. We used a batch size of 64 and the Adam optimizer (Kingma & Ba, 2014) with a decaying learning rate initialized to 0.001 and weight decay set to 0.00005. We also included a diffusion version of the proposed AR framework, also using the GatedGCN architecture. For this model, we follow the general conventions of DIFUSCO, using discrete diffusion with cosine schedule. We used a batch size of 256 and the Adam optimizer (Kingma & Ba, 2014) with decaying learning rate initialized to 0.0002 and weight decay set to 0.00005. For both models, we also applied layer normalization, residual connections, and dropout (0.2). For the exact hyperparameters, please refer to Appendix F.

---

[3]TSP– indicates TSP instances containing – nodes.

## 5.1 Experimental Results

The main results are presented in Table 1. The two baselines demonstrated strong performance on TSP instances of the same size used in training (Tsp50), especially Difusco. This is expected, as Difusco has been shown as a state-of-the-art NCO method (Sun & Yang, 2024). In comparison, our AR framework slightly outperforms both EfficientTsp and Difusco on the Tsp50 test set, achieving this with fewer model parameters and less training data. Furthermore, the NAR models show substantial performance degradation as they attempt to generalize beyond the training instance size. In fact, their optimality gaps drop from 2.91% and 0.79% on Tsp50 to 30.38% and 13.14% on Tsp100 for EfficientTsp and Difusco respectively. Our model, on the other hand, achieved superior performance across all test sets with larger problem instances (Tsp100, Tsp200, Tsp500), displaying a stronger ability to generalize.

Table 1: Results against existing GNN-based supervised methods. All models are trained and validated on Tsp50 instances.

| Algorithm | Problem Instances | Parameters | Gap % ↓ | | | |
|---|---|---|---|---|---|---|
| | | | Tsp50 | Tsp100 | Tsp200 | Tsp500 |
| EfficientTsp Nar | 10,000 | 33mil | 2.91 | 30.38 | 37.58 | 51.21 |
| Difusco Nar | 1,502,000 | 5.3mil | 0.79 | 13.14 | 18.89 | 32.95 |
| | 10,000 | | 11.92 | 21.62 | 36.84 | 54.28 |
| Ours Ar | 10,000 | 3.5mil | **0.65** | **3.9** | **10.75** | **17.95** |
| Ours (Diffusion) Ar | 10,000 | 4mil | 12.86 | 14.89 | 15.91 | 18.16 |

Notably, when Difusco is limited to 10,000 training instances, similar to our models, it experiences a marked decline in performance across all problem sizes. On the Tsp50 dataset, Difusco's optimality gap drops from a near-optimal 0.79% to 11.92%. These results suggest that Difusco may require very large training sets in order to perform well. Our framework, by contrast, achieves slightly better optimality gap than Difusco on same-size test instances with significantly less training data. Finally, even when limited to 10,000 training instances, the diffusion version of our proposed AR framework displayed comparable results with Difusco on Tsp50 and superior performance on Tsp100, Tsp200, and Tsp500. In fact, on Tsp200 and Tsp500, its performance even surpasses that of Difusco when trained on 1,502,000 training instances. This supports the hypothesis that the superior performance of our proposed framework can be attributed to its autoregressive nature. For comparisons with other NCO methods that are non-supervised or non-GNN-based, as well as results of the proposed model trained using more instances, please refer to Appendix J.

## 6 Conclusion

In this paper, we examined the general supervised NAR solution construction framework across three well-known combinatorial optimization problems and six GNN architectures. We identified a misalignment between the desired optimization objective (optimality gap) and the surrogate optimization objective (training loss). To address this, we proposed a general supervised AR framework that leverages conditional generation. Our training process involves sampling partial solutions from optimal solutions and training the model to complete them. Empirical results show that our proposed framework does not exhibit the previously observed misalignment and leads to improved performance. Notably, when compared against existing GNN-based supervised methods on TSP datasets of various sizes, our AR framework displays superior performance, especially in terms of generalizing to larger instances.

### 6.1 Limitations and Future Work

One promising extension of our work is the development of more sophisticated encoding mechanisms for the current partial solutions, possibly tailored to specific combinatorial optimization problems. The iterative nature of the AR methods in general, while beneficial for solution quality, incurs computational costs. Future research could focus on exploring AR approaches that balance solution quality with computational efficiency. Lastly, our approach only uses one optimal solution per problem instance for training while combinatorial optimizations can have several different optimal

solutions. Future research could try to further capture the multi-modal nature of combinatorial optimization problems by incorporating all optimal solutions of any given problem instance.

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

## A   Formal Problem Definitions

In this section, we formally define the three combinatorial optimization problems studied in this paper: Traveling Salesperson Problem (TSP), Maximum Independent Set (MIS), and Minimum Vertex Cover (MVC). These three problems are well-known NP-complete combinatorial optimization problems (Karp, 2010). They are common canonical examples of computational intractability and many real-world applications reduce to these formulations.

### A.1   Traveling Salesperson Problem

The Traveling Salesperson Problem involves finding the shortest possible route that visits each city exactly once and returns to the starting point. Formally, let $\mathcal{G} = (V, E)$ be a complete, weighted graph, where $V$ is the set of nodes representing cities, and $E$ is the set of edges representing the paths between them. Each edge $(u, v) \in E$ is associated with a non-negative weight $w(u, v)$, representing the distance between nodes $u$ and $v$. The objective is to find a Hamiltonian cycle $C \subseteq E$ that minimizes the total travel cost, expressed as:

$$\min \sum_{(u,v) \in C} w(u, v) \tag{10}$$

subject to visiting each vertex $v \in V$ exactly once (Karp, 2010; Lawler, 1985). The TSP has numerous applications in logistics, manufacturing, and planning, making it a pivotal problem in both theoretical and practical optimization research.

### A.2   Maximum Independent Set

The Maximum Independent Set problem involves finding the largest set of mutually non-adjacent nodes in a graph. Formally, given an undirected graph $\mathcal{G} = (V, E)$, an independent set $I \subseteq V$ is a set of nodes such that no two nodes in $I$ are connected by an edge in $E$. The objective is to maximize the size of such an independent set (Karp, 2010; Lewis, 1983):

$$\max |I| \tag{11}$$

subject to:

$$\forall u, v \in I, \quad (u, v) \notin E. \tag{12}$$

The MIS problem has applications in network theory, social network analysis, and computational biology, where identifying the largest group of mutually independent entities is often of interest.

### A.3 MINIMUM VERTEX COVER

The Minimum Vertex Cover problem involves identifying the smallest subset of nodes that collectively cover all edges of a given graph. Formally, let $\mathcal{G} = (V, E)$ be an undirected graph. A vertex cover $S \subseteq V$ is defined as a subset of nodes such that each edge $(u, v) \in E$ is incident to at least one vertex in $S$. The objective is to find a vertex cover of minimum cardinality (Karp, 2010; Lewis, 1983):

$$\min |S| \tag{13}$$

subject to:

$$\forall (u, v) \in E, \quad u \in S \text{ or } v \in S. \tag{14}$$

The MVC problem is of critical importance in network security, resource allocation, and bioinformatics, where covering critical connections with minimal resources is often required.

## B EXTENDED RELATED WORK

In Section 1, we categorized existing NCO methods into supervised learning, unsupervised learning, and reinforcement learning, and discussed the supervised learning framework in detail. In this section, we provide a brief overview of the unsupervised learning and reinforcement learning methods, categorized into *methods for routing problems* (i.e., TSP and related vehicle routing problems (VRP)) and *methods on other graph-based combinatorial optimization problems*.

### B.1 UNSUPERVISED LEARNING

Recent works within unsupervised learning for NCO predominantly employ NAR methods. The goal is to develop models that can learn combinatorial optimization solutions without labeled data, utilizing techniques like probabilistic modeling and objective relaxation.

**Methods for Routing Problems.** Min et al. (2024); Min & Gomes (2024) propose unsupervised methods for the TSP, introducing approaches that generate edge probability maps and leverage the Gumbel-Sinkhorn operator for permutation representation.

**Methods for Other Graph-Based Problems.** Toenshoff et al. (2021) introduce a generic GNN architecture for maximum constraint satisfaction problems, training unsupervised on small instances to effectively solve larger ones. Karalias & Loukas (2020) employ a probabilistic method for CO, creating a framework that finds integral solutions via neural network parametrization over sets. Amizadeh et al. (2019) propose a neural framework for solving the Circuit-SAT problem through an unsupervised differentiable approach. Wang et al. (2022) present a relaxation-based approach for CO with neural networks, particularly effective for applications without explicitly defined objectives. Schuetz et al. (2022) use a physics-inspired GNN model to solve CO problems framed as quadratic unconstrained binary optimizations, achieving scalability and strong performance.

### B.2 DEEP REINFORCEMENT LEARNING

Deep Reinforcement Learning (DRL) methods construct solutions iteratively through learning-based policies, optimizing for long-term rewards. While DRL methods achieved strong performance on

a range of graph-based combinatorial optimization problems, they require a significant amount of computational resources and takes a long time to converge.

**Methods for Routing Problems.** Kool et al. (2019) use attention mechanisms and REINFORCE training to solve routing problems like TSP and VRP, achieving near-optimal results. Xing & Tu (2020) combine GNNs with MCTS to tackle TSP, outperforming recent learning-based methods. Peng et al. (2020) introduce a dynamic attention model with an encoder-decoder architecture. Zhou et al. (2023) propose a meta-learning framework for VRP generalization across varying sizes and distributions, while Gao et al. (2023) design an ensemble policy with a local transferable policy to boost generalization across different distributions and scales.

For large-scale routing problems, Son et al. (2024) develop Equity-Transformer, using Transformer architecture to solve min-max routing problems efficiently across large instances. Luttmann & Xie (2024) propose a neural method tailored for picker routing in mixed-shelves warehouses, developing a novel encoder and hierarchical decoding scheme for CO on heterogeneous graphs. Bresson & Laurent (2021) adapt Transformer networks to solve TSP using reinforcement learning and beam search, improving upon learned heuristics with minimal optimality gaps.

**Methods for Other Graph-Based Problems.** Outside of routing, several DRL works focus on different combinatorial tasks. Drori et al. (2020) propose a GNN-based reinforcement learning framework to solve general CO problems in linear time, covering diverse graph types. Ahn et al. (2020) introduce a DRL scheme for MIS, dynamically adjusting solution stages to improve scalability on large graphs. Kwon et al. (2021) present MatNet for matrix-form CO problems, showing efficacy for asymmetric TSP and flexible Flow Shop Scheduling. Zhang et al. (2020) apply deep reinforcement learning to Job Shop Scheduling, using GNNs for robust policy network representation and achieving strong performance on unseen large instances. Qiu et al. (2022) address the scalability of CO problems with DIMES, leveraging a compact continuous space and meta-learning for efficient training. Finally, Sanokowski et al. (2023) introduce variational annealing for CO, using subgraph tokenization to enhance performance on complex problem instances.

Unsupervised learning methods in NCO focus on learning solution structures without labels, often using probabilistic models and objective relaxations. DRL approaches, on the other hand, optimize policies iteratively for solution construction, showing particular strength in diverse problems but requiring huge computational resources.

## C RELATION TO GUIDED TREE SEARCH

Guided Tree Search (Li et al., 2018) is a supervised method for constructing solutions to the MIS problem. It employs a tree search algorithm that leverages probability maps generated by a Graph Convolutional Network (GCN) (Kipf & Welling, 2017). The training process is the same to that of the NAR framework we examine in Section 3, with a a slight difference: the GCN produces multiple probability maps for each graph instance in a single forward pass. Specifically, each node is assigned scores $m$ times (with $m = m$ in their configuration), representing its likelihood of inclusion in 32 different potential solutions. This approach acknowledges the presence of multiple optimal solutions for each MIS instance, allowing the model to generate a diverse set of focused probability maps.

The GCN is trained using a hindsight cross-entropy loss:

$$\mathcal{L}_{\text{hindsight}} = \min_{i=1,...,m} \mathcal{L}_{\text{CE}}\left(y_{\text{true}}, f_i\right) \tag{15}$$

where $\mathcal{L}_{\text{CE}}$ is the standard cross-entropy loss, $y_{\text{true}}$ are the true labels, and $f_1, ..., f_m$ are the $m$ different probability maps produced by the GCN. The training aims to minimize the lowest cross-entropy loss among all generated maps. A tree search algorithm is then employed to construct a feasible solution from these maps, switching between different maps as needed.

During the traversal of a single probability map, the method iteratively includes the node with the highest probability into the solution and marks all its neighbors as excluded. After this step, all marked nodes, both included and excluded, are removed from the graph. The GCN then generates a new probability map for the remaining subgraph. This process repeats until all nodes have been marked.

In Section 2, we classify this method as non-autoregressive. However, it differs from other NAR methods because it conceptually resembles conditional generation in autoregressive methods. To condition the probabilities on the existing partial solution, this method does not explicitly train the GCN to generate conditioned outputs. Instead, it reduces the original graph after each selection by removing the selected node and its neighbors, and feeds this reduced subgraph back into the GCN to produce a new probability map. This is a unique property of the MIS problem, where the state of the current partial solution can be implicitly represented by the modified input graph.

Formally, given a graph $\mathcal{G} = (V, E)$ and an independent set of nodes $K \subseteq V$, let $K' = K \cup \bigcup_{v \in K} N(v)$, where $N(v)$ denotes the set of neighbors of node $v$. Let $\mathcal{G}' = \mathcal{G}[V \setminus K']$ be the induced subgraph obtained by removing $K'$ from $\mathcal{G}$. If $L$ is a maximum independent set on $\mathcal{G}'$, then $K \cup L$ constitutes the largest independent set on $\mathcal{G}$ that includes $K$.

In other words, finding the largest independent set on $\mathcal{G}$ conditioned on the partial solution $K$ is equivalent to finding the maximum independent set on the reduced subgraph $\mathcal{G}'$. In $\mathcal{G}'$, all nodes in $K$ and their neighbors, along with all edges incident to them, have been removed. This reduction effectively encodes the condition into the graph structure.

This property suggests that while the model does not explicitly perform conditional generation, it achieves a similar outcome by manipulating the input graph to reflect the current partial solution. Moreover, this approach of implicit conditional generation through input manipulation does not readily extend to other combinatorial optimization problems. As such, we classify this method as non-autoregressive because the GCN model itself is not trained to generate predictions autoregressively.

## D  DATASET GENERATION

To generate the MVC and MIS instances used in this study, we generate Erdős–Rényi (ER) graphs (Erdos et al., 1960) using the NetworkX library (Hagberg et al., 2008). Each $n$ node ER graph is generated with edges established between node pairs with probability $p = 0.05 \pm 0.02$.

For TSP, we generate Waxman graphs (Waxman, 1988) using NetworkX Hagberg et al. (2008) with the parameter $\beta = 1$ and domain coordinates ranging from $(0, 0)$ to $(1, 1)$. In this setup, each node is assigned a coordinate within the unit square, resulting in a complete graph where every pair of nodes is connected. Edge costs are calculated based on the Euclidean distance between node coordinates. To make the TSP graphs more computationally tractable, we sparsify them by connecting each node only to its $k = 20$ nearest neighbors. This graph generation process, including sparsification, follows the conventions established in prior TSP studies (Joshi et al., 2019; Sun & Yang, 2024; Fu et al., 2021).

To generate the ground-truth labels, we obtain an optimal solution for each problem instance using Gurobi (Gurobi Optimization, LLC, 2023).

## E  DECODING ALGORITHMS

In this section, we detail the greedy search algorithms for each of the three problems (TSP, MIS, MVC) within both the existing NAR frameworks and the proposed AR framework.

### E.1  TRAVELING SALESPERSON PROBLEM

In the NAR framework, given a probability map over all edges, we start by randomly selecting a node as the current position. We then iteratively construct the tour by greedily choosing the incident edge with the highest probability from the current node. After selecting an edge, we update the current node to the adjacent node connected by that edge. This process repeats, with already visited nodes being masked to prevent revisiting, until a valid TSP tour is completed. This is the greedy decoding method used in EFFICIENTTSP (Joshi et al., 2019) and is similar to the variant in DIFUSCO (Sun & Yang, 2024), where edge probabilities are adjusted by dividing them by their corresponding edge costs.

In our proposed AR framework, we remove the constraint of sequential tour construction. Starting with an empty solution set and a probability distribution over all edges, we greedily add the edge with

the highest probability to the solution set, provided it does not violate any TSP constraints and is not already included. After each addition, we use the trained GNN model to generate a new probability distribution based on the updated solution set. This iterative process continues until a valid TSP tour is formed. We found that the non-sequential variant of the search algorithm yields slightly better performance.

## E.2 MAXIMUM INDEPENDENT SET

In the NAR framework, we begin with a probability map over all nodes. We iteratively select the node with the highest probability that is neither already selected nor adjacent to any previously selected nodes. This process repeats until every node is either selected or adjacent to a selected node.

In the AR framework, we follow the same process but generate a new probability map after each node selection.

## E.3 MINIMUM VERTEX COVER

In the NAR framework, we start with a probability map over all nodes and iteratively select the highest-probability node that has not yet been chosen. This continues until all edges are covered—that is, every edge has at least one endpoint in the solution set.

In the AR framework, we follow the same process but generate a new probability map after each node selection.

# F MODEL CONFIGURATIONS

In this section, we detail the configurations and hyperparameters of the models used in our experiments. All models were implemented using the Deep Graph Library (DGL) (Wang et al., 2019). For definitions and additional information on specific hyperparameters, please refer to the DGL documentation. All diffusion models in this paper follow the configurations used in the implementation of DIFUSCO (Sun & Yang, 2024).

## F.1 NON-AUTOREGRESSIVE MODELS

This subsection outlines the configurations for the NAR models discussed in Section 3.

The following hyperparameters are consistent across all NAR models:

- Batch size: 64
- Number of layers: 4
- Hidden dimension: 128
- Batch normalization: Enabled
- Residual connections: Enabled

Specific settings for each model architecture are as follows:

- GAT: Number of heads = 2
- GatedGCN: N/A
- GCN: N/A
- GIN: Apply function layers = 2; Learn $\epsilon$ = True; Aggregation type = Max
- MoNet: Aggregation type = Max; Pseudo-dimension = 2; Number of kernels = 1
- GraphSage: Aggregation type = Pool

We use the Adam optimizer (Kingma & Ba, 2014) with the following parameters:

- Initial learning rate: 0.001

- Learning rate reduction factor: 0.5
- Patience: 3 epochs
- Weight decay: 0.00005
- Number of epochs: 20

## F.2 PROPOSED AUTOREGRESSIVE MODELS

This subsection provides the configurations for the AR models presented in Section 4 and Section 5.

The following hyperparameters are shared across all AR models in Section 4:

- Batch size: 64
- Number of layers: 4
- Hidden dimension: 128
- Batch normalization: Enabled
- Residual connections: Enabled
- Number of partial solutions sampled: 50

Specific configurations for each AR model architecture in Section 4 are:

- GAT: Number of heads = 2
- GatedGCN: N/A
- GCN: N/A
- GIN: Apply function layers = 2; Learn $\epsilon$ = True; Aggregation type = Max
- MoNet: Aggregation type = Max; Pseudo-dimension = 2; Number of kernels = 1
- GraphSage: Aggregation type = Pool

The optimizer settings are identical to those used in the NAR models:

- Optimizer: Adam (Kingma & Ba, 2014)
- Initial learning rate: 0.001
- Learning rate reduction factor: 0.5
- Patience: 3 epochs
- Weight decay: 0.00005
- Number of epochs: 20

For the proposed AR model applied to the TSP discussed in Section 5, we use the following configuration:

- Architecture: GatedGCN
- Number of layers: 8
- Hidden dimension: 256
- Layer normalization: Enabled
- Residual connections: Enabled
- Dropout: 0.2
- Number of partial solutions sampled: 200

The optimizer settings for this model are:

- Optimizer: Adam (Kingma & Ba, 2014)
- Initial learning rate: 0.001

- Learning rate reduction factor: 0.5
- Patience: 3 epochs
- Weight decay: 0.00005
- Number of epochs: 50

# G    RUNTIME ANALYSIS

In this section, we analyze the runtime performance of the proposed AR framework compared to the NAR framework. As expected, the AR framework requires more time to construct solutions due to the additional inference steps involved. Table 2 presents the experimental results on the average runtime per problem instance for all models used in the paper.

Firstly, for the models in Sections 3 and 4, the results indicate that the AR models experiences an average runtime increase of around 388% for TSP, 81% for MIS, and 69% for the MVC compared to the NAR models. The higher increase for TSP is attributed to the difference in decoding as we did not construct the TSP tour sequentially in the AR models whereas they were constructed sequentially in the NAR models. Detailed descriptions of the search algorithms can be found in Appendix E. This approach required checking more edges in each iteration, in addition to performing more inference steps.

Table 2: Average runtime (in seconds) per problem instance. The NAR row refers to the models studied in Section 3 and the AR row refers to the models studied in Section 4. The last three rows refer to the models studied in Section 5. All experiments performed on the same hardware.

| MODELS | | PROBLEM SIZE | RUNTIME S ↓ | | |
| --- | --- | --- | --- | --- | --- |
| | | | TSP | MIS | MVC |
| NAR | SECTION 3 | TSP100 | 8.8 | 0.31 | 0.16 |
| AR | SECTION 4 | | 42.9 | 0.56 | 0.27 |
| DIFUSCO | | | 0.86 | - | - |
| EFFICIENTTSP | SECTION 5 | TSP50 | 0.36 | - | - |
| OURS | | | 5.1 | - | - |

For the models in Section 5, the proposed AR model showed an average runtime increase of around one order of magntidue when compared against EFFICIENTTSP and DIFUSCO. However, these comparisons are not too precise as there can be significant runtime differences depending on the exact implementation details of the models and the search algorithms.

It is worth noting that the AR framework can be adjusted to add multiple nodes or edges per iteration, although this modification was not implemented in this paper. Such an adjustment would introduce a trade-off between solution quality and runtime efficiency, which could be explored in future work.

# H    PSEUDOCODE FOR TRAINING PROCESS OF PROPOSED AR FRAMEWORK

Algorithm 1 is the pseudocode for the training process of the proposed framework.

# I    DISCUSSION ON BASELINES

In this section, we discuss the baselines used in Section 5, specifically DIFUSCO (Sun & Yang, 2024) and EFFICIENTTSP (Joshi et al., 2019). We explain the slight discrepancies in the performance of DI-FUSCO compared to its original paper and justify the use of reported performance for EFFICIENTTSP rather than reimplementing it ourselves.

**DIFUSCO**    We found an error in the implementation of DIFUSCO in the authors' public repository[4], where the solution improvement technique 2-OPT was applied, even when disabled in the configura-

---

[4]https://github.com/Edward-Sun/DIFUSCO

---

**Algorithm 1** Training Process for Proposed AR Framework

---

**Require:** Training instances $\{g_i, \hat{x}_i\}_{i=1}^N$, where $g_i = (V_i, E_i)$ is a graph and $\hat{x}_i$ is the set of variables true in optimal solution
**Ensure:** Trained GNN Model $\mathcal{M}$
 1: Initialize empty training set $T \leftarrow \emptyset$
 2: **for** each training instance $(g_i, \hat{x}_i)$ **do**
 3:     Sample a partial solution $\tilde{x}_i \subset \hat{x}_i$ with size uniformly sampled from $\{1, ..., |\hat{x}_i|\}$
 4:     Define the remainder of the solution $\tilde{x}_i' = \hat{x}_i \setminus \tilde{x}_i$
 5:     **for** each decision variable $v \in g_i$ **do**
 6:         **if** $v \in \tilde{x}_i$ **then**
 7:             Set binary feature $f_{g_i}(v) \leftarrow 1$                     $\triangleright$ Indicates inclusion in $\tilde{x}_i$
 8:         **else**
 9:             Set binary feature $f_{g_i}(v) \leftarrow 0$
10:         **end if**
11:         **if** $v \in \tilde{x}_i'$ **then**
12:             Assign label $l_{g_i}(v) \leftarrow 1$                     $\triangleright$ Indicates $v$ should be selected next
13:         **else**
14:             Assign label $l_{g_i}(v) \leftarrow 0$
15:         **end if**
16:     **end for**
17:     Update training set $T \leftarrow T \cup (g_i, \{f_{g_i}(v)\}_{v \in G_i}, \{l_{g_i}(v)\}_{v \in G_i})$
18:     Repeat the above process $k$ times to sample $k$ partial solutions
19: **end for**
20: Train model $\mathcal{M}$ on training set $T$ by minimizing the cross-entropy loss
21: **return** Trained GNN Model $\mathcal{M}$

---

tion. After fixing this, their results on TSP50 as reported in Section 5 are slightly worse than those reported in their original manuscript, despite using the same checkpoint.

**EFFICIENTTSP**    The original repository for EFFICIENTTSP is no longer functional, so we could not access the provided checkpoints. Instead, we used an alternative implementation from a public repository[5] maintained by the same authors. Despite following the configuration described in the original manuscript, we were unable to reproduce the reported results on TSP50. Therefore, we relied on the results reported from their original manuscript for the TSP50 results.

## J    COMPARISON WITH NON-GNN NEURAL COMBINATORIAL OPTIMIZATION METHODS

We compare our proposed method with other NCO solution construction approaches that are not based on GNNs using TSP50 instances as the primary benchmark. Consistent with our methodology in Section 5, we include only studies that employ greedy decoding strategies. As these approaches do not utilize GNNs, they fall outside the scope of the paper and therefore are not included in our analysis. We report these results in the appendix for completeness.

The proposed model used in this evaluation was trained following the same procedure outlined in Section 5, but with 200,000 problem instances instead of 10,000. Please note that the baseline results are sourced from their original publications and have not been obtained on identical test datasets. Some methods are excluded from this comparison because they do not report results on TSP50 instances. The results are presented in Table 3.

Our proposed framework outperforms all included baselines except for the DRL transformer-based method introduced by Bresson & Laurent (2021). However, we achieve performance comparable to the transformer network. These findings demonstrate that our method is competitive with leading NCO solution construction methods for the TSP in general.

---

[5]https://github.com/chaitjo/learning-tsp

Table 3: Results against existing NCO methods that utilize greedy search. All models are trained on TSP50 instances. All baseline results are sourced from their original manuscripts.

| ALGORITHM | TSP50 GAP % ↓ |
|---|---|
| OURS SL | *0.35* |
| IMAGE DIFFUSION SL (GRAIKOS ET AL., 2022) | 1.28 |
| POINTER NETWORK SL (VINYALS ET AL., 2015) | 11.4 |
| TRANSFORMER NETWORK DRL (BRESSON & LAURENT, 2021) | **0.31** |
| AM DRL (KOOL ET AL., 2019) | 1.76 |
| NCORL DRL (BELLO ET AL., 2016) | 4.54 |
| POMO DRL (KWON ET AL., 2020) | 0.64 |
| EAN DRL (DEUDON ET AL., 2018) | 2.23 |
| S2VDQN DRL (KHALIL ET AL., 2017) | 5.81 |

