# OpenReview forum: "What's Wrong With Non-Autoregressive Graph Neural Networks in Neural Combinatorial Optimization"
_ICLR.cc/2025/Conference — ICLR 2025 Conference Withdrawn Submission_

### Official Review · Reviewer_5Tmo · 2024-11-02

**Soundness:** 3
**Presentation:** 4
**Contribution:** 3
**Rating:** 8
**Confidence:** 2

**Summary:**

This paper identifies a critical limitation in current supervised non-autoregressive GNN approaches for neural combinatorial optimization, where a misalignment exists between the training loss and the optimality gap. The problem is illustrated by testing various GNN architectures across multiple combinatorial optimization tasks, revealing that a lower training loss does not necessarily result in better solutions. The authors attribute this issue to the use of a single probability map, which does not align well with search algorithms that greedily build solutions. To address this, they propose an autoregressive approach in which GNNs are trained to complete partial solutions, producing probability maps conditioned on the current solution state. Training is accomplished by sampling subsets of optimal solutions. Empirical results indicate that this approach successfully mitigates the misalignment and outperforms existing GNN-based methods, generalizing to larger problem instances.

**Strengths:**

1. The paper addresses a significant issue of misalignment between training loss and optimality gap, which is common in existing supervised GNN-based approaches.
2. The proposed autoregressive GNN approach, which completes partial solutions and can correct suboptimal decisions made earlier, is innovative, effective, and well-suited.
3. The empirical results are promising, with the first experiment validating that the method addresses the misalignment issue and the second demonstrating superior performance over other GNN-based methods.
4. The writing is clear and well-structured, with strong motivations and a clearly explained methodology.

**Weaknesses:**

1. In the experiments where the misalignment problem is observed, only a greedy search algorithm is used, where at each step, one variable is added to the solution. This brings the question of whether the problem is universal, i.e. if the misalignment still occur with other search algorithms. For example, if the search algorithm is not iterative in nature (a simple example is setting the threshold of 0.5 to determine if a solution should be included), whether the misalignment is still a problem.
2. The greedy search algorithm adds one variable at a time. However, from my understanding, the GNN learns to complete a partial solution, not having one variable change at a time. This seems to misalign with the conditional probability desired by the search algorithm.
3. What is the multimodal nature of combinatorial optimization that the author mentions that the proposed method better captures?

**Questions:**

See weakness.

---

> ### Comment · Reviewer_SVvL · 2024-11-18
>
> Dear Reviewer 5Tmo,
>
> As a researcher with several years of experience in NCO, I find the manuscript's writing clarity and comparison with AR methods inadequate. I have detailed my concerns in my review.
> After reading both my comments and Reviewer ZQRH's review, I would appreciate your could share your opinion about this issues.

---

> > ### Comment · Reviewer_5Tmo · 2024-11-18
> >
> > Hi, thank you for initiating the discussion. I have read both your and Reviewer ZQRH's reviews, and I agree with the concerns you raised. However, I was waiting for the rebuttals from the authors, after which I will re-evaluate my rating. I believe it would be fairer to also hear what the authors say. Thank you!

---

> ### Author Response · Authors · 2024-11-20
> **Reply to Reviewer 5Tmo**
>
> Thank you for your detailed review and high rating of our submission. We deeply appreciate your time and we are confident we can address your concerns.
>
> ---
>
> W1/Q1: We appreciate the reviewer’s suggestion to explore additional search algorithms. Advanced algorithms, such as beam search or sampling, can utilize additional computational resources to explore a broader solution space and therefore compensate for lower-quality predictions by the GNN. In fact, prior studies have demonstrated that sophisticated search algorithms, such as Monte Carlo Tree Search [1] and Guided Tree Search [2], can achieve near-SOTA results when relying on suboptimal or even random probability maps. By employing greedy search in our experiments, we aim to isolate and evaluate the quality of the GNN’s predictions and their alignment with the optimization objective.
>
> For non-iterative search algorithms, such as a threshold-based approach (e.g., setting a threshold of 0.5), feasibility is not inherently guaranteed. This is due to the lack of mechanisms that ensure all the variables above the threshold do not violate the problem’s constraints.
>
> We appreciate the reviewer’s interest in other search algorithms. As such, we are currently extending our analysis to include other search algorithms. Preliminary results indicate that the misalignment remains an issue when using sampling (5 samples) as the search algorithm: https://ibb.co/sPSTRZR
>
> ---
>
> W2/Q2: We appreciate the reviewer’s observation regarding the interaction between GNN predictions and the search process. The GNN is trained to predict the most likely variables that best complete the current partial solution, producing a probability distribution conditioned on the partial solution at each step. In greedy search, this predicted distribution is used to select the next variable to be added to the partial solution. Crucially, after each selection, the partial solution changes, which alters the conditions for subsequent decisions. To maintain consistency between the conditional probability map and the evolving partial solution, the GNN computes an updated probability map to reflect the new conditions. For example, in the Maximum Independent Set (MIS) problem, selecting a node affects the feasibility of its neighboring nodes being included in the solution. By dynamically updating the probability map, the GNN adapts to these changes, ensuring alignment with the current state of the partial solution. For a more detailed example on the importance of conditional generation, the reviewer is referred to the illustrative example we provided in our response to Reviewer SVvL (Q1).
>
> ---
>
> W3/Q3: Thank you for raising this question on the multimodal nature of CO problems. Many CO problems feature multiple distinct optimal solutions. For example, in the MIS problem, several structurally different solutions may be equally optimal. NAR methods, which construct solutions based on a single, fixed probability map, often face challenges in these scenarios. They can become trapped between several optimal solutions, predicting nodes that belong to different and conflicting optimal solutions. This results in variables being selected based on a probability map that inadequately represents the evolving partial solution, leading to lower solution quality.
>
> In contrast, our AR framework dynamically updates the probability map based on the current partial solution. This iterative conditional generation allows the model to adapt to the evolving context, avoiding conflicts caused by competing modes. As a result, our method better captures the multimodal nature of CO problems, where multiple conflicting paths to optimality often coexist. For a detailed example, we refer the reviewer to the illustrative example provided to Reviewer SVvl (Q1).
>
> ---
>
> We’d like to once again thank the reviewer’s thoughtful comments. We hope these clarifications and additional experiments comprehensively address the reviewer’s concerns. We welcome any further questions you may have.
>
> ---
>
> References
>
> [1] Yifan Xia, Xianliang Yang, Zichuan Liu, Zhihao Liu, Lei Song, and Jiang Bian. Position: Rethinking post-hoc search-based neural approaches for solving large-scale traveling salesman problems. In Forty-first International Conference on Machine Learning, 2024.
>
> [2] Maximilian Böther, Otto Kißig, Martin Taraz, Sarel Cohen, Karen Seidel, and Tobias Friedrich. What’s wrong with deep learning in tree search for combinatorial optimization. In International Conference on Learning Representations, 2022

---

> > ### Comment · Reviewer_5Tmo · 2024-11-25
> >
> > Thank you for the detailed responses. All my questions are well answered. I also appreciate the authors for additional experiments. I believe the paper explores a critical question, which itself is an important contribution. Therefore, I chose to maintain my score. However, after reading the comments from other reviewers, I agree with the questions on (i) solution being an AR approach (ii) insufficient data. I chose to lower my confidence score to reflect that.

---

> > > ### Author Response · Authors · 2024-11-27
> > > **Reply to Reviewer 5Tmo**
> > >
> > > Thank you for your response and high evaluation of our work. We truly appreciate your feedback. We also wanted to clarify that we’ve addressed the two points that you pointed in our responses to the other reviewers:
> > >
> > > >(i) solution being an AR approach
> > >
> > > We describe our motivation behind proposing an AR approach in our response to Reviewer SVvl (see Q1 in the initial response as well as our follow-up comments).
> > >
> > > >(ii) insufficient data
> > >
> > > We provide additional results demonstrating the misalignment with 1M training data for the two main baselines (see Figure 2 in the updated manuscript, also available here https://ibb.co/qp4R10y) as well as demonstrating the lack of misalignment with 200K training data for the proposed method (available here: https://ibb.co/dcMgDH0). Furthermore, we provide additional comparative analysis with 1M training data for the baselines and 200K training data for the proposed method, available here: https://ibb.co/5Yb2hm8. We include further clarification and context regarding the training sets in our most recent response to Reviewer ZQRH.

---

### Official Review · Reviewer_ZQRH · 2024-11-04

**Soundness:** 1
**Presentation:** 2
**Contribution:** 2
**Rating:** 5
**Confidence:** 4

**Summary:**

This paper investigates the supervised Non-Autoregressive Graph Neural Network frameworks and finds a misalignment between the training objective (i.e., minimizing the loss function) and the quality of the constructed solutions.
In summary : improvements in the quality of the probability maps do not necessarily lead to better solutions.
The reason authors provide is that NAR approaches assume independence between different variables, ignoring the dependencies present in combinatorial optimization problems.

The paper proposes using autoregressive (AR) models with supervised learning. They propose to consider the previously predicted variables when predicting the next one, capturing the conditional dependencies. The proposed framework propose to perform training to complete partial solution.

The authors empirical show that their proposed method outperforms Non Autoregressive methods on 3 problems(TSP, MVC and MIS) . Further it also generalizes better to larger size data.

Concerns:
I have some concerns related to amount of training data used and usage of only greedy decoding to perform benchmarking of existing works. Please see weakness.

**Strengths:**

1. The idea of using Supervised learning(SL) and auto-regressive approach of CO is relevant.
2. Making prediction based upon partial solution seems to be novel for NCO and SL.

**Weaknesses:**

1. For training, we used 10,000 instances of TSP503 for our model. For the baselines,
1,502,000 instances of TSP50 are used to train DIFUSCO and 10,000 instances of TSP50 are used to
train EFFICIENTTSP, as per their original manuscripts.

-> is 10000 a large number? previous studies have used millions of instances[A].
Why was it limited to only 10000? How does the performance change with increase in dataset size ( for different methods).
Does the conclusion remain same. I would expect to see training/validation curves for different problems and methods.



2. "For each problem, the training set consists of 5,000 random synthetically generated
problem instances, each with 100 nodes"
For explaining failure of NR supervised learning methods the authors use 5000 training samples. It seems to be a very small number.  Check [A]. I would recommend the authors to use a significantly large training dataset.

3. "we employ greedy search in order to evaluate the impact of the probability maps in isolation."

Why only greedy for all problems, what restricted the authors not to use ideas like beam-search/ sampling for all problems, if not for TSP ? Check [A]. Can we conclude with just greedy decoding?



[A] An Efficient Graph Convolutional Network Technique
for the Travelling Salesman Problem
https://arxiv.org/pdf/1906.01227

**Questions:**

Check weakness.  I am majorly concerned by the amount of training data used for the experiments.

---

> ### Author Response · Authors · 2024-11-20
> **Reply to Reviewer ZQRH**
>
> Thank you for your detailed review and highlighting the relevance and novelty of our approach. We deeply appreciate your time and we are confident we can address your concerns.
>
> ---
>
> W1/Q1: For the referenced experiment, we used 10,000 training instances per epoch to ensure a fair comparison with one of our baselines, which uses this dataset size in its original configuration. We consider achieving superior results without requiring a large dataset as a benefit of the proposed method, as it reduces both training and labeling costs.
>
> However, we understand the reviewer’s interest in the impact of training set size. To address this, we conducted additional experiments with larger training datasets during the rebuttal period. In these experiments, we increased the number of training instances from 10,000 to 1,000,000 for the EfficientTSP baseline and from 10,000 to 200,000 for our proposed method (DIFUSCO was excluded as it was already trained on over 1,000,000 instances). We are not currently able to complete experiments using 1,000,000 training instances for our proposed method during this short time but we will complete them later and update the manuscript with the results. The experimental results for TSP50 are shown here: https://ibb.co/L94PCCL
>
> These results further support the conclusions made in our manuscript as our proposed method outperforms all GNN-based supervised approaches. Furthermore, the improvement on TSP50 from using additional data brings our method to a level competitive with the state-of-the-art for NCO methods on TSP, including non-GNN and non-supervised approaches. The results for the larger TSP instances will require more time to compute but will be included in our revised manuscript. We greatly appreciate the reviewer’s suggestion, as it has allowed us to further substantiate our evaluation.
>
> ---
>
> W2/Q2: We understand the reviewer’s concern regarding the training set size used in our analysis of NAR methods. As shown in Figure 2 (page 6), we demonstrate the misalignment in two NAR methods that were trained on larger datasets: EfficientTSP [1] and DIFUSCO [2], trained on 10,000 and 1,502,000 TSP instances, based on their respective original papers.
>
> To further validate the observed misalignment, we have re-conducted these experiments using more training instances, training EfficientTSP on 1,000,000 instances of TSP50. The results confirm that the misalignment persists, supporting the original conclusions. We will include these results in the revised manuscrip: https://ibb.co/SPVPX6B [Edit: fixed typo in graph.]
>
> While we were unable to rerun all NAR model experiments with larger training sets during the rebuttal period due to computational constraints, we emphasize that the additional NAR models presented in the paper are generalizations or extensions of the EfficientTSP framework which, as demonstrated, exhibits the misalignment even when trained on a significantly larger dataset. We will include these additional experiments in the final manuscript.
>
> ---
>
> W3/Q3: We appreciate the reviewer’s suggestion to explore additional search algorithms. Advanced algorithms, such as beam search or sampling, can utilize additional computational resources to explore a broader solution space and therefore compensate for lower-quality probability maps. In fact, prior studies have demonstrated that sophisticated search algorithms, such as Monte Carlo Tree Search [1] and Guided Tree Search [2], can achieve near-SOTA results when relying on suboptimal or even random probability maps:
>
> * Xia et al. [3] demonstrated that Monte Carlo Tree Search achieves near-SOTA results on TSP using probability maps generated by a distance-based softmax heuristic.
> * Bother et al. [4] showed that Guided Tree Search [5] achieves near-optimal results on MIS using completely random probability maps.
>
> To isolate the impact of the GNN predictions, we opted to use greedy decoding in our experiments. This choice highlights the misalignment between the quality of the generated probability maps and the optimality gap.
>
> To address the reviewer’s interest in other search algorithms, we are currently extending our experiments. At this moment, we have preliminary results from re-running the experiments using a sampling procedure with 5 samples for the DIFUSCO model showing that the misalignment persists: https://ibb.co/sPSTRZR
>
> ---
>
> We'd like to once again thank the reviewer's thoughtful comments. We hope these clarifications and the additional experiments comprehensively address the reviewer’s concerns. Your feedback has significantly strengthened the soundness of our work, and we are happy to address any further questions you may have.

---

> > ### Author Response · Authors · 2024-11-20
> > **References**
> >
> > [1] Chaitanya K Joshi, Thomas Laurent, and Xavier Bresson. An efficient graph convolutional network technique for the travelling salesman problem. arXiv preprint arXiv:1906.01227, 2019.
> >
> > [2] Zhiqing Sun and Yiming Yang. Difusco: Graph-based diffusion solvers for combinatorial optimization. Advances in Neural Information Processing Systems, 36, 2024.
> >
> > [3] Yifan Xia, Xianliang Yang, Zichuan Liu, Zhihao Liu, Lei Song, and Jiang Bian. Position: Rethinking post-hoc search-based neural approaches for solving large-scale traveling salesman problems. In Forty-first International Conference on Machine Learning, 2024.
> >
> > [4] Maximilian Böther, Otto Kißig, Martin Taraz, Sarel Cohen, Karen Seidel, and Tobias Friedrich. What’s wrong with deep learning in tree search for combinatorial optimization. In International Conference on Learning Representations, 2022
> >
> > [5] Zhuwen Li, Qifeng Chen, and Vladlen Koltun. Combinatorial optimization with graph convolutional networks and guided tree search. Advances in neural information processing systems, 31, 2018.

---

> > ### Comment · Reviewer_ZQRH · 2024-11-24
> > **Thanks**
> >
> > Thanks for rebuttal. But I believe unless all major experiments are conducted with large number of training samples( as in existing studies), a clean conclusion cannot be made. Example figure 4.
> >
> > Or if the authors can show any study which shows the number of samples are used by them( for all methods) are sufficiently large and adding more samples won't change the conclusion.
> >
> > I have updated my score.

---

> > > ### Author Response · Authors · 2024-11-27
> > > **Reply to Reviewer ZQRH**
> > >
> > > Thank you for your response and for updating your score. We truly appreciate your feedback.
> > >
> > > We would like to further clarify that the major results—specifically, the results for the proposed method and the two main baselines on TSP (EfficientTSP [1] and DIFUSCO [2])—have been produced using large training datasets. This includes:
> > >
> > > 1. The analysis of misalignment for TSP50: DIFUSCO was trained on 1.5M samples, matching the training set size used in the original DIFUSCO paper [2]. EfficientTSP was trained on a dataset of 1M samples, scaled up from the 10,000 training instances used in its original manuscript at the reviewer's request. The results are shown in Figure 2 (Section 3) of the updated manuscript, also available here: https://ibb.co/qp4R10y. Additionally, we trained the proposed method on a scaled-up training set consisting of 200K instances, results available here: https://ibb.co/dcMgDH0. All the additional results provided support the original conclusions made in the manuscript.
> > >
> > > 2. The performance in the main comparative analysis: All models were trained on TSP50 and evaluated on TSP50, TSP100, TSP200, and TSP500. The results for these experiments have been included here (as before, DIFUSCO is trained on its original 1.5M samples, EfficientTSP uses the scaled-up training set of 1M samples, the proposed method uses the scaled-up training set of 200K samples): https://ibb.co/5Yb2hm8.
> > >
> > > While we have not yet scaled up experiments for all GNN architectures to 1M training instances (due to the short rebuttal period), these are not published benchmarks (to the best of our knowledge). They are variants of EfficientTSP that we introduced to demonstrate that the patterns observed in EfficientTSP hold across GNN architectures and problem settings. Nonetheless, we plan to include these scaled-up experiments for the variants in the final manuscript.
> > >
> > > Additionally, we would like to note that while many NCO methods do utilize training sets of 1M instances, several top methods have achieved their performance using significantly smaller datasets. For example, in our extended analysis results shown in Table 3 (Appendix J) of the revised manuscript, also available here: https://ibb.co/RGFfvVR:
> > >
> > > * Transformer-Network [3]: The *best-performing method* in our extended analysis uses *10K training instances* per epoch (similar to our previous, smaller-scale experiments).
> > >
> > > * POMO [4]: The *second-best-performing baseline* in our extended analysis uses *100K training instances* per epoch.
> > >
> > > ---
> > >
> > > References
> > >
> > > [1] Chaitanya K Joshi, Thomas Laurent, and Xavier Bresson. An efficient graph convolutional network technique for the travelling salesman problem. arXiv preprint arXiv:1906.01227, 2019.
> > >
> > > [2] Zhiqing Sun and Yiming Yang. Difusco: Graph-based diffusion solvers for combinatorial optimization. Advances in Neural Information Processing Systems, 36, 2023.
> > >
> > > [3] Bresson, Xavier, and Thomas Laurent. "The transformer network for the traveling salesman problem." arXiv preprint arXiv:2103.03012 (2021).
> > >
> > > [4] Kwon, Yeong-Dae, et al. "Pomo: Policy optimization with multiple optima for reinforcement learning." Advances in Neural Information Processing Systems 33 (2020): 21188-21198.

---

> ### Comment · Reviewer_SVvL · 2024-11-25
>
> I want to point out that the conclusion
> > prior studies have demonstrated that sophisticated search algorithms, such as Monte Carlo Tree Search [1] and Guided Tree Search [2], can achieve near-SOTA results when relying on suboptimal or even random probability maps
>
> only holds for small size TSPs, which is easier to search for optimal solutions. In other words, it is misleading since this conclusion cannot extend to the whole TSP context. In fact, as shown in Xia et al. [3], generating better heatmap help the search process to find better solution.
>
> I disagree with the authors' excessive claims about the search process's dominance, as it also depends on the quality of heatmaps generated by NAR.

---

> ### Author Response · Authors · 2024-11-27
> **Reply to Reviewer SVvl**
>
> Thank you for your comment regarding advanced search algorithms.
>
> ---
>
> > I want to point out that the conclusion:
> [prior studies have demonstrated that sophisticated search algorithms, such as Monte Carlo Tree Search [1] and Guided Tree Search [2], can achieve near-SOTA results when relying on suboptimal or even random probability maps]
> only holds for small size TSPs, which is easier to search for optimal solutions. In other words, it is misleading since this conclusion cannot extend to the whole TSP context. In fact, as shown in Xia et al. [3], generating better heatmap help the search process to find better solution. I disagree with the authors' excessive claims about the search process's dominance, as it also depends on the quality of heatmaps generated by NAR.
>
> Firstly, Xia et al. demonstrated that, using MCTS, their simple distance-based heuristic can outperform NCO methods (that also use MCTS) on large problem instances, up to TSP-10000 [1].
>
> Secondly, we agree with the reviewer on the importance of the quality of the generated probability maps and their role in guiding the search algorithm to find high-quality solutions. This is precisely why we use greedy search. As noted in our initial response to Reviewer ZQRH, we employ greedy search to decouple the impact of probability map quality from that of more sophisticated search procedures. Sophisticated search algorithms, such as MCTS, can obscure the evaluation of probability map quality even on large-scale problem instances [1]. By isolating this factor, we can more effectively assess whether the models are generating high-quality probability maps.
>
> While there may be learning approaches designed specifically for a particular search algorithm (and not generalizable to others), this lies beyond the scope of our investigation. We will clarify this distinction in the paper. Similarly, we do not compare with algorithms that learn branching heuristics for exact solvers.
>
> Thank you again for your additional comments.
>
> ---
>
> References
>
> [1] Yifan Xia, Xianliang Yang, Zichuan Liu, Zhihao Liu, Lei Song, and Jiang Bian. Position: Rethinking post-hoc search-based neural approaches for solving large-scale traveling salesman problems. In Forty-first International Conference on Machine Learning, 2024.

---

> ### Comment · Reviewer_SVvL · 2024-12-02
>
> First, Xia et al. [1] does not outperform all the NAR baselines in their results, and it is not the SOTA (State of the Art) of NAR in recent months.
>
> Secondly, no application of TSP solely relies on heat-maps without post-searching. It is a toy setting. If the authors want to demonstrate the applicability of their algorithm, they can compare their results based on the same post-searching process in more practical and popular settings. Additionally, please note that there is no conflict between these two experiments; authors can still retain the experiments with heat-maps only as an ablation study.
>
> I do not think it is reasonable to simplify other methods and introduce a comparison under narrow setting.

---

> > ### Author Response · Authors · 2024-12-04
> > **Reply to Reviewer SVvL**
> >
> > Thank you for your comment.
> >
> > Xia et al. demonstrated that, using MCTS, a probability map generated by a simple heuristic can produce solutions of comparable quality to those generated by trained GNNs [1]. This reference was provided as evidence that advanced search algorithms can be used to compensate for lower-quality probability maps and not as a SOTA baseline.
> >
> > [1] Xia, Yifan, et al. "Position: Rethinking Post-Hoc Search-Based Neural Approaches for Solving Large-Scale Traveling Salesman Problems." Forty-first International Conference on Machine Learning, 2024.

---

### Official Review · Reviewer_SVvL · 2024-11-04

**Soundness:** 2
**Presentation:** 2
**Contribution:** 1
**Rating:** 3
**Confidence:** 5

**Summary:**

This paper investigates non-autoregressive (NAR) Graph Neural Networks (GNNs) for neural combinatorial optimization and identifies a key misalignment between training loss and solution quality. The authors attribute this misalignment to NAR models’ inability to capture inter-variable dependencies, and they propose an autoregressive (AR) alternative to improve the alignment between training objectives and solution quality. Empirical evaluations on standard problems such as Traveling Salesperson Problem (TSP), Maximum Independent Set (MIS), and Minimum Vertex Cover (MVC) demonstrate that the AR approach outperforms existing supervised GNN-based NAR methods, particularly in generalizing to larger instances.

**Strengths:**

This work stands out for its empirical examination of various GNN architectures and the experimental insights into the limitations of NAR models on the combinatorial optimization tasks TSP, MIS, and MVC. The proposed AR framework, which iteratively builds solutions based on partial solutions, shows promising improvements over traditional NAR approaches. Furthermore, the authors provide a comprehensive experimental setup and benchmarks, which enhances the clarity of the paper and the credibility of its results.

**Weaknesses:**

While the paper identifies valid shortcomings in NAR methods, it does not fully address these limitations within the NAR framework itself. Instead of proposing an enhancement that would make NAR models more effective, the authors shift to an AR approach, which is already well-established for combinatorial optimization tasks. Given that current state-of-the-art AR models, particularly in the TSP domain, achieve near-optimal solutions on problems up to TSP200, the choice to compare a new AR model to NAR models could be seen as a misalignment of goals. Although the empirical results favor the AR model over NAR approaches, this comparison does not substantially advance the field of NAR or show a path forward for improving NAR’s inherent limitations. Moreover, the results on the AR framework fall short of reaching or challenging the current state-of-the-art in AR models, which diminishes the work’s impact in terms of advancing solution quality or generalization ability for AR approaches.

**Questions:**

Can the authors clarify why an autoregressive method was chosen instead of attempting to refine or address the limitations of NAR methods directly? It would be helpful to understand whether any NAR-specific modifications were considered.

Given that state-of-the-art AR approaches already perform well for small and medium-sized TSP instances, how does the proposed AR method compare to these benchmarks in terms of optimality and scalability? The AR framework here still falls short of achieving performance competitive with top-performing AR models, which raises the question of its comparative effectiveness.

---

> ### Author Response · Authors · 2024-11-20
> **Reply to Reviewer SVvl (Part 1)**
>
> Thank you for your detailed review and your high evaluation of our empirical examination and comprehensive experiment set up. We deeply appreciate your time and we are confident we can address your concerns.
>
> ---
>
> Q1: We thank the reviewer for raising this important question. Our work highlights a fundamental issue of misalignment in existing NAR supervised GNN-based methods. We reason that this misalignment arises from the static nature of the probability maps generated by NAR models, which are unable to capture the multimodal nature of CO problems. We do not believe this can be easily resolved within the NAR framework, as its reliance on a single predicted probability map hinders its ability to capture the combinatorial dependencies between variables and limits the model's ability to adapt to changes in the partial solution throughout the solution construction process.
>
> To illustrate this, consider the following example of the Maximum Independent Set (MIS) problem on a simple 8-node graph (https://ibb.co/X7nxHb8). The green nodes indicate one optimal solution with an objective value of 4, while the yellow nodes indicate a different optimal solution, also with an objective value of 4. Evidently, each node is part of some optimal solution, and as such, a NAR method may output similar probabilities for all nodes. This behavior aligns with its intended purpose, which is to output the likelihood of each node being part of an optimal solution. As all probabilities are the same, suppose the search algorithm selects nodes 1 and then 7. Since no additional feasible nodes can be added, the algorithm terminates, resulting in a solution of size 2, which is far from optimal. The issue arises because, while both nodes 1 and 7 belong to optimal solutions, they do not belong to the same optimal solution. The inclusion of both nodes restricts the algorithm from reaching an optimal solution due to feasibility constraints.
> Now consider an AR model capable of conditional generation. The AR model may generate a similar probability map in the first iteration where all nodes have similar probabilities. However, suppose the algorithm selects node 1 first, the AR model would update the probability map conditioned on the fact that node 1 has already been selected. This conditional update would allow the model to predict nodes that are more likely to be part of the same optimal solution as node 1 (e.g., nodes 6 and 3), potentially avoiding the selection of nodes that, while part of some optimal solution, are incompatible with node 1 (e.g., node 7). Following the example, the AR model would select node 6, update the probability map again to condition on nodes 1 and 6, and then select node 3. The algorithm would continue iterating until no feasible nodes remain. This process would result in a better solution than the NAR method in this example. The issue with NAR methods as illustrated by this example is consistent with the observed misalignment because improving the quality of the single probability map does not consistently lead to improved solutions.
>
> As a result of this fundamental misalignment of supervised GNN-based NAR approaches, we decided to focus on developing a robust AR framework as an alternative approach that does account for the dependencies and can update the probability map accordingly. In contrast, we don't believe the existing NAR framework that relies on a single predicted probability map can overcome this issue.

---

> > ### Author Response · Authors · 2024-11-20
> > **Reply to Reviewer SVvl (Part 2)**
> >
> > Q2: We thank the reviewer for raising this point. While numerous AR methods exist, supervised GNN-based AR methods remain largely unexplored. Despite being widely studied in recent literature, prior research on supervised GNN-based methods for CO problems has predominantly focused on NAR frameworks. Our investigation specifically addresses supervised GNN-based methods and we believe that identifying and addressing the limitations of NAR methods is critical for advancing this field, and our work represents a significant step in that direction. Our results highlight AR supervised GNN-based methods as a promising alternative that addresses the observed limitations of NAR approaches, which we hope will inspire further research in this underexplored area.
> >
> > Still, we acknowledge the importance of comparison with state-of-the-art. To address this, we conducted additional experiments during the rebuttal period with increased training datasets (e.g., 200,000 training graphs per epoch). We observe that our approach is competitive with the best-performing Transformer-based model. We also note that while many AR methods are specifically tailored to TSP, our framework is designed to generalize across a broader range of CO problems (as we demonstrate in the paper for MIS and MVC). These updated results will  be included in our revised manuscript: https://ibb.co/GRrLgwV
> >
> > ---
> >
> > We'd like to once again thank the reviewer's thoughtful comments. Your questions have allowed us to clarify key aspects of our work. We hope these clarifications and additional experiments highlight the significance of our contributions, and we welcome any further questions you may have.

---

> ### Comment · Reviewer_SVvL · 2024-11-24
>
> In my opinion, the title of this manuscript is too general and do not match exactly what authors found. Even though authors found that the misalignment under there setting: improvements in the quality of probability maps do not correlate with higher quality of constructed solutions.
>
> However, as Reviewer ZQRH concerns, only 5000 training data usually cause overfitting (loss reduce) and hinder further solving the problem.
>
> Moreover, I think there lacks theoretical proof to say that "there is wrong in NAR". Diffusco generate high quality solution and recent SOTA [1] in diffusion and heatmap also shows promising results in NAR. I think there still space to improve the performance of NAR. This manuscript only provide some experimental results under unfair settings. The content of this manuscript is not enough to say "there is wrong in NAR" , at least before authors prove that there is not hope for L2O methods to learn to generate a optimal heatmap (the probabilities of optimal nodes in each row is 1).
>
> [1] Fast T2T: Optimization Consistency Speeds up Diffusion-based Training to Testing for Combinatorial Optimization

---

> ### Comment · Reviewer_SVvL · 2024-11-25
>
> First, theoretically, an instance should typically provide the full information for solving NCO. Both NAR and AR can use this information to find the optimal solution. The first paragraph of Q1 retells the characteristics of AR but does not highlight the novelty of this work. As illustrated in the original paper on DIFFUSCO, they achieved a 0.00% gap in TSP 50 and TSP 100. However, this work removes the tree search process of DIFFUSCO in the experiment (gap increases to 0.79%), and the authors claim they outperform (with gaps still large at 0.65% and 3.9%). Note that even for TSP-10000, DIFFUSCO achieved a 2.58% gap, which is significantly smaller than the 3.9% achieved by this work for TSP-100.
>
> Second, in Q2, the author only compares very old AR methods, which are from 3-7 years ago and are no longer state-of-the-art. Especially, one of the key contributions of POMO is using augmentation and sampling multiple trajectories (with gaps of 0.03% for TSP 50 and 0.14% for TSP 100). However, in the results provided by the authors, they remove these from POMO and only sample a single trajectory (causing the gap to increase to 0.64%), and claim their method outperforms POMO—a method proposed four years ago. They also apply the same weakening approach to other baselines.
>
> Third, when asked to compare to AR methods, the authors not only oversimplify the baselines but also underplay the fact that they focus on supervised GNN without any evidence to support that supervised GNN has a certain advantage in solving NCO, in order to avoid comparing to the most SOTA AR methods.
>
> In conclusion:
>
> For the theoretical part, 1) this work **only shows a phenomenon** of a mismatch in loss and gap for NAR, but fails to contribute to the rationale behind this. 2) Limiting the context to a certain architecture does not necessarily mean contributing to the literature, whether it is GNN or Transformer.
>
> For the performance part, this work removes critical parts for both NAR (post-search) and AR (sampling), and the only conclusion is that this supervised GNN greedily outperforms simplified selected baselines **in such a narrow setting**. It fails to provide a broad impact in building a new SOTA for more popular and common settings. Without further justifying the specific advantages of limiting to greedy search and supervised GNN, I cannot see a solid contribution to the literature.
>
> I will not increase my score, after carefully reading the rebuttals to all the reviewers.

---

> ### Author Response · Authors · 2024-11-27
> **Reply to Reviewer SVvl (Part 1)**
>
> Thank you for your response. In the following, we address the points you raised:
>
> ---
>
> > only 5000 training data usually cause overfitting (loss reduce) and hinder further solving the problem
>
> In our updated manuscript, we test the two main baselines (DIFUSCO and EfficientTSP) with over 1,000,000 training instances and observe similar patterns (see the detailed response to Reviewer ZQRH and Figure 2 in the updated manuscript). These experiments match the scale of training data used in DIFUSCO's original paper [1].
>
> ---
>
> > Diffusco generate high quality solution and recent SOTA [1] in diffusion and heatmap also shows promising results in NAR. I think there still space to improve the performance of NAR.
>
> We acknowledge that NAR methods, particularly diffusion-based models, can achieve impressive empirical results, and we do not claim that there is no room for improving NAR methods. However, we highlight an important phenomenon in these models: the misalignment between the training objective and the optimization objective.
>
> Regarding the title, it was inspired by prior work in the field [2]. If the reviewer finds the title too general, we are open to revising it to more concretely emphasize the observed misalignment.
>
> ---
>
> > This manuscript only provide some experimental results under unfair settings.
>
> The results reported in the paper in Figure 2 and Table 1 are based on the same configurations used in prior work. We have not altered any experimental settings from their original papers, including the training data.
>
> ---
>
> > at least before authors prove that there is not hope for L2O methods to learn to generate a optimal heatmap (the probabilities of optimal nodes in each row is 1).
>
> Thank you for this suggestion. We believe that is precisely what we demonstrated in the illustrative example provided in our previous comment. The referenced example is a counterexample that shows that even with a perfect NAR method that accurately predicts the participation of each variable within optimal solutions, it is not possible to produce a single probability map where "the probabilities of optimal nodes in each row is 1." This is because there are multiple optimal solutions, and all nodes can participate in one. We showed, using this counterexample, that a perfect NAR method will highlight all nodes in the graph as equally optimal therefore providing no useful guidance for solving the problem. In contrast, a perfect AR method updates the probability map at each step. Initially, the predictions may align with those of the NAR method, but once a node is chosen, subsequent probability maps are conditioned on that choice. The probabilities for nodes that can no longer participate in an optimal solution containing the current partial solution will be reduced.
>
> Given the reviewer's emphasis on this aspect, we will include a discussion of this concept, along with the illustrative counterexample, in the final manuscript to further motivate our AR approach.
>
> ---
>
> Part 1/2

---

> ### Author Response · Authors · 2024-11-27
> **Reply to Reviewer SVvl (Part 2)**
>
> > For the theoretical part, 1) this work only shows a phenomenon of a mismatch in loss and gap for NAR, but fails to contribute to the rationale behind this. 2) Limiting the context to a certain architecture does not necessarily mean contributing to the literature, whether it is GNN or Transformer.
>
> Please refer to our previous paragraph (in Part 1 of this reply) regarding our insight into the observed misalignment.
> Furthermore, we do not claim that supervised GNNs possess specific advantages over other approaches, such as deep reinforcement learning (DRL) methods. Rather, we believe it is crucial to study and understand these different approaches. Our contribution lies in providing insight into an important issue inherent to supervised GNN-based methods—an approach that is popular in the NCO domain and has achieved impressive empirical results [1].
>
> ---
>
> > For the performance part, this work removes critical parts for both NAR (post-search) and AR (sampling), and the only conclusion is that this supervised GNN greedily outperforms simplified selected baselines in such a narrow setting. It fails to provide a broad impact in building a new SOTA for more popular and common settings. Without further justifying the specific advantages of limiting to greedy search and supervised GNN, I cannot see a solid contribution to the literature.
>
> Please see our response to your comment under Reviewer ZQRH’s review.
>
> ---
>
> > Note that even for TSP-10000, DIFUSCO achieved a 2.58% gap, which is significantly smaller than the 3.9% achieved by this work for TSP-100.
>
> The results presented in our paper are generalization results (i.e., models trained on TSP50 instances and tested on TSP100/200/500 instances.) while the results the reviewer is referring to are achieved by a model trained on TSP10000 instances. We refer the reader to Section 5 in our paper for detailed description of our experimental setting.
>
> ---
>
> > the author only compares very old AR methods, which are from 2–7 years ago and are no longer state-of-the-art
>
> The methods included in our extended analysis on TSP-50 (Table 3) are well-established baselines still used in recent NCO papers, such as DIFUSCO (2023) [1] and the work of Xia et al. (2024) [3]. Furthermore, since these approaches are not GNN-based, this comparison primarily serves to demonstrate that our approach achieves "strong" results that are comparable to SOTA methods and is not inherent to our investigation of GNN-based approaches.
>
> ---
>
> Thank you again for your additional comments.
>
> ---
>
> References
>
> [1] Zhiqing Sun and Yiming Yang. Difusco: Graph-based diffusion solvers for combinatorial optimization. Advances in Neural Information Processing Systems, 36, 2023.
>
> [2] Maximilian Böther, Otto Kißig, Martin Taraz, Sarel Cohen, Karen Seidel, and Tobias Friedrich. What’s wrong with deep learning in tree search for combinatorial optimization. In International Conference on Learning Representations, 2022.
>
> [3] Yifan Xia, Xianliang Yang, Zichuan Liu, Zhihao Liu, Lei Song, and Jiang Bian. Position: Rethinking post-hoc search-based neural approaches for solving large-scale traveling salesman problems. In Forty-first International Conference on Machine Learning, 2024.
>
> ---
>
> Part 2/2

---

> ### Comment · Reviewer_SVvL · 2024-12-02
>
> Testing DIFUSCO without post-searching does not provide impactful results. Besides, EfficientTSP was the first GNN method to solve TSP, but it was proposed five years ago. Almost all the methods in recent papers can outperform it. We should not overemphasize outperforming this method.
>
> What I claimed is that the models "highlight an important phenomenon: the misalignment between the training objective and the optimization objective," but they fail to provide insightful analysis and instead turn to other paradigms, overshadowing the contribution. The paper "What’s Wrong with Deep Learning in Tree Search for Combinatorial Optimiziation" focuses on tree search and provides much analysis and experiments for tree search. However, this work simply show there is a phenomenon in NAR; it is not sufficient to justify the title 'What's Wrong With Non-Autoregressive'. The authors should carefully consider the title before being "inspired" by other papers' titles.
>
> What the authors compared for NAR and AR is incorrect. They assume NAR can only decode all solutions at the same time, while AR decodes only one node at a time. In fact, a step of AR softmax is equivalent to a row of the heat-map of NAR. They both represent probabilities for all nodes. We can decode probabilities and decide on the next node in any possible way.
>
> Finally, does training on TSP50 include any advantage in a popular computational setting? Most recent methods train on instances of 100 or 200 and then generalize to larger ones. Why not test them in the same setting? I suspect that the main reason is that the training of this GNN-based method is too slow due to computational overhead, and it fails to train on larger datasets (only 200k) and on larger sizes of TSP within a reasonable time frame, even over a rebuttal period of two weeks.

---

> > ### Author Response · Authors · 2024-12-04
> > **Reply to Reviewer SVvL**
> >
> > Thank you for your comments. We believe we have addressed the main points in our previous responses. To summarize:
> >
> > * We provided a clear example illustrating the limitation of using a single probability map in NAR methods, where the predicted probabilities can offer poor guidance independent of how the probabilities are decoded. Specifically, we showed an example where all nodes have similar probabilities of participating in optimal solutions. This inherent limitation of NAR models that rely on a single probability map motivated our investigation into AR approaches, which do not suffer from this drawback. We believe this is a valid motivation and that our investigation provides valuable insights into both AR and NAR models. We do not claim that NAR methods cannot be further improved in various ways.
> >
> > * Our focus on greedy search is driven by our interest in isolating the impact of probability map quality. Evidence in the literature suggests that sophisticated search algorithms can trade off additional computational effort to compensate for lower-quality probability maps. While we acknowledge the importance of search algorithms and do not claim otherwise, our primary goal is to investigate the misalignment concerning the generated probability maps. Greedy search was chosen for this purpose. While we are currently running experiments with additional search algorithms and will include these results in the manuscript, we believe the results based on greedy search are important and provide meaningful insights into a limitation of current NAR approaches that use a single, static probability map.
> >
> > * As mentioned previously, we are willing to revise the title if the reviewer finds it too “strong.”
> >
> > * We will run the requested scaled-up experiments and include the results in the manuscript.

---

### Author Response · Authors · 2024-11-23
**Rebuttal Revisions**

We would like to thank the reviewers for their thoughtful and detailed comments. Below, we detail the revisions made to the manuscript:

* Corrected typos, improved phrasing, and adjusted formatting.
* Added experimental results for the EfficientTSP model trained on additional problem instances to **Figure 2** (Section 3).
* Updated **Table 3** (Appendix J) with revised results for the proposed model trained on additional problem instances.
* Revised the experimental setup section to reflect the newly conducted experiments.

---

Once again, we would like to thank the reviewers for their discussion and feedback which has greatly strengthened our manuscript.

---

### Note · Authors · 2025-01-12

**Comment:**

We thank the reviewers for their thoughtful feedback. After consideration, we have decided to withdraw the paper to address these suggestions more thoroughly.

**Withdrawal Confirmation:**

I have read and agree with the venue's withdrawal policy on behalf of myself and my co-authors.